# Pro-inflammatory hepatic macrophages generate ROS through NADPH oxidase 2 via endocytosis of monomeric TLR4–MD2 complex

So Yeon Kim [1], Jong-Min Jeong[1], Soo Jin Kim[2], Wonhyo Seo[1], Myung-Ho Kim[2], Won-Mook Choi[2], Wonbeak Yoo[2], Jun-Hee Lee[2], Young-Ri Shim[1], Hyon-Seung Yi[3], Young-Sun Lee [4], Hyuk Soo Eun[3], Byung Seok Lee[3], Kwangsik Chun[5], Suk-Jo Kang[6], Sun Chang Kim[6,7], Bin Gao[8], George Kunos[8], Ho Min Kim[2] & Won-Il Jeong[1,2]

Reactive oxygen species (ROS) contribute to the development of non-alcoholic fatty liver disease. ROS generation by infiltrating macrophages involves multiple mechanisms, including Toll-like receptor 4 (TLR4)-mediated NADPH oxidase (NOX) activation. Here, we show that palmitate-stimulated CD11b$^+$F4/80$^{low}$ hepatic infiltrating macrophages, but not CD11b$^+$F4/80$^{high}$ Kupffer cells, generate ROS via dynamin-mediated endocytosis of TLR4 and NOX2, independently from MyD88 and TRIF. We demonstrate that differently from LPS-mediated dimerization of the TLR4–MD2 complex, palmitate binds a monomeric TLR4–MD2 complex that triggers endocytosis, ROS generation and increases pro-interleukin-1β expression in macrophages. Palmitate-induced ROS generation in human CD68$^{low}$CD14$^{high}$ macrophages is strongly suppressed by inhibition of dynamin. Furthermore, Nox2-deficient mice are protected against high-fat diet-induced hepatic steatosis and insulin resistance. Therefore, endocytosis of TLR4 and NOX2 into macrophages might be a novel therapeutic target for non-alcoholic fatty liver disease.

[1] Laboratory of Liver Research, Biomedical Science and Engineering Interdisciplinary Program, KAIST, Daejeon 34141, Republic of Korea. [2] Graduate School of Medical Science and Engineering, KAIST, Daejeon 34141, Republic of Korea. [3] Department of Internal Medicine, Chungnam National University School of Medicine, Daejeon 35015, Republic of Korea. [4] Department of Internal Medicine, Korea University College of Medicine, Seoul 08308, Republic of Korea. [5] Department of Surgery, Chungnam National University School of Medicine, Daejeon 35015, Republic of Korea. [6] Department of Biological Sciences, Korea Advanced Institute of Science and Technology, Daejeon 34141, Republic of Korea. [7] Intelligent Synthetic Biology Center, 373–1, Guseong-dong, Yuseong-gu, Daejeon 34141, Republic of Korea. [8] National Institutes of Health (NIH), National Institute on Alcohol Abuse and Alcoholism (NIAAA), Bethesda, MD 20892, USA. Correspondence and requests for materials should be addressed to W.-I.J. (email: wijeong@kaist.ac.kr)

Diet-induced obesity is commonly associated with non-alcoholic fatty liver disease (NAFLD) and insulin resistance, in which recruited immune cells such as macrophages, neutrophils, lymphocytes, mast cells and eosinophils contribute to a pro-inflammatory environment[1, 2]. Of these cells, recruited macrophages and resident Kupffer cells secrete diverse cytokines including tumor necrosis factor-α (TNF-α), interleukin-6 (IL-6) and IL-1β, which contribute to insulin resistance in hepatocytes by upregulating the activation of inhibitor of κB kinase-β/nuclear factor-κB (NF-κB) and c-Jun N-terminal kinase (JNK)/activator protein-1[2–4]. In this process, increased plasma levels of saturated free fatty acids (FFAs), such as palmitic acids, triggers pro-inflammatory signals through Toll-like receptor 2 (TLR2) and TLR4 in macropahges[5].

Reactive oxygen species (ROS) are one of the major causes in steatohepatitis and insulin resistance. Infiltrated macrophages generate ROS by multiple mechanisms, including mitochondria damage, endoplasmic reticulum stress, and nicotinamide adenine dinucleotide phosphate (NADPH) oxidases (NOXs)[6, 7]. Among members of the NOX family, NOX2, also known as gp91phox, is the phagocytic NADPH oxidase, constitutively associated with p22phox in plasma membrane. Upon activation of NOX2, phosphorylated p47phox is translocated to p22phox, recruiting the additional components p67phox, p40phox and Rac GTPases to assemble a complex with NOX2, which then generates superoxide[8]. In liver resident Kupffer cells and infiltrating macrophages, NOX2-derived ROS stimulates them to produce pro-inflammatory cytokines such as TNF-α, IL-6, and IL-1β, in response to various factors including oxidized low-density lipoprotein (LDL) and lipopolysaccharide (LPS)[9–11]. In contrast to classically activated inflammatory macrophages, termed as M1, alternatively activated macrophages (M2) by Th2 cytokines IL-4 and IL-13 secrete anti-inflammatory cytokine IL-10, which protects from inflammation-mediated insulin resistance by inhibiting the deleterious effects of pro-inflammatory cytokines on insulin signaling[3]. A phenotype switch from M1 to M2, associated with reduced ROS formation, is brought about by the absence NOX2 due to its diminished mRNA stability or genetic ablation[11, 12]. Until now, the exact roles of NOX2 in inducing steatosis and insulin resistance in hepatic macrophages have not been clearly understood.

TLR4 plays a key role for innate immune responses and its signaling is triggered by the transfer of its ligand LPS to a TLR4–MD2 complex, which then undergoes homodimerization[13]. This homodimer then can activate MyD88-dependent or MyD88-independent pathways. The MyD88-dependent pathway is activated at the plasma membrane, whereas the TRIF-dependent (MyD88-independent) pathway requires internalization of TLR4 into endosome in a dynamin-dependent manner[14, 15]. In addition to LPS, TLR4 can bind a wide range of ligands including FFAs[2, 3, 5]. The mechanisms underlying TLR4 binding to diverse agonist are not clear, although the heterodimerization of TLR4 or the involvements of co-receptors and accessory proteins such as CD14, CD36 and MD2 have been suggested for the recognition of diverse TLR4 agonists[14]. Nonetheless, LPS-mediated or FFA-mediated TLR4 signaling activates the same downstream transcriptional factors such as NF-κB and interferon-regulatory factors to induce inflammatory responses[14]. Recently, a line of studies has suggested that sensing of LPS by TLR4 mediates priming and activation of several NOX family members, leading to enhanced inflammatory responses including ROS generation, the NLR family pyrin domain containing 3 (NLRP3) inflammasome-associated IL-1β production, and NF-κB-mediated IL-6 production in mouse macrophages and human blood monocytes[16–18]. While the obesity-related increase in circulating FFA promotes the development of hepatic steatosis and insulin resistance, the precise molecular mechanisms involved in the interplay among FFA, TLR4 and NOX2 have not been fully understood.

Here, we report that in CD11b⁺F4/80low hepatic macrophages, palmitate triggers the endocytosis of a monomeric TLR4–MD2 complex, leading to NOX2 activation and ROS generation, which play an obligatory role in diet-induced hepatic steatosis and insulin resistance. These findings provide novel pathophysiological insights and therapeutic targets for NAFLD.

## Results

**NOX2 deficiency attenuates HFD-induced hepatic steatosis.** To investigate the role of NOX2 in vivo, wild type (*WT*) and *Nox2* KO mice were fed a high-fat diet (HFD) for 12 weeks. At sacrifice, body, liver and epididymal fat pad weights were significantly lower in *Nox2* KO than *WT* mice, whereas food intake was similar in the two strains (Fig. 1a, b). On gross observation, livers of *Nox2* KO mice were less yellowish in color than those of *WT* mice (Fig. 1b). Serum levels of alanine aminotransferase (ALT), aspartate aminotransferase (AST), triglyceride (TG) and total cholesterol (TC) were significantly lower in *Nox2* KO than *WT* mice (Fig. 1c). *Nox2* KO mice had less liver fat, as reflected in hepatic TG levels, and higher liver glycogen content than *WT* mice (Fig. 1d, e). Furthermore, protein levels of sterol regulatory element-binding protein-1 (SREBP1) and fatty acid synthase (FAS) were remarkably reduced, whereas levels of phosphorylated AMP kinase (AMPK) and AKT were significantly increased in *Nox2* KO compared to *WT* mice, indicating increased insulin sensitivity in the former (Fig. 1f). Consistently, basal blood glucose was lower, and glucose tolerance and insulin sensitivity were higher in *Nox2* KO than in *WT* mice, as quantified by glucose tolerance test (GTT) and insulin tolerance test (ITT), respectively (Fig. 1g). These differences were evident as early as 6 weeks of HFD feeding (Supplementary Fig. 1). These data suggest that NOX2 might be correlated with hepatic steatosis and insulin resistance in HFD-fed mice.

**NOX2 deficiency decreases ROS production in CD11b⁺F4/80low cells.** Next, FACS analyses revealed that hepatic populations of lymphocytes such as NK cells (NK1.1⁺CD3⁻), NKT cells (NK1.1⁺CD3⁺), CD4 T cells (CD4⁺CD3⁺), CD8 T cells (CD8⁺CD3⁺), and regulatory T cells (Treg; CD4⁺CD25⁺Foxp3⁺) were similar between *WT* and *Nox2* KO mice (Fig. 2a). However, the population of infiltrated CD11b⁺F4/80low macrophages was significantly decreased in the absence of NOX2, while Ly6G⁺CD11b⁺ neutrophils and CD11b⁺F4/80high Kupffer cells showed no significant differences in livers of *Nox2* KO and *WT* mice (Fig. 2b). ROS generation was remarkably decreased in CD11b⁺F4/80low macrophages of *Nox2* KO compared to *WT* mice, whereas CD11b⁺F4/80high Kupffer cells produced similar level of ROS (Fig. 2c), suggesting the dominant role of CD11b⁺F4/80low macrophages in ROS generation in response to HFD feeding. In parallel with these findings, quantitative RT–PCR (qRT–PCR) analyses showed that the expression of *Tnf-α* and *Il-1β* was significantly lower in liver mononuclear cells (MNCs) of *Nox2* KO mice compared to *WT* mice (Fig. 2d). Moreover, immunoblots of liver tissues revealed that phosphorylated NF-κB and JNK proteins were significantly reduced (Fig. 2e), which was paralleled by a significant reduction in apoptotic cells in *Nox2* KO compared to *WT* (Fig. 2f). However, in vitro exposure to palmitate did not induce apoptosis of CD11b⁺F4/80low macrophages from either *WT* or *Nox2* KO mice (Fig. 2g). Collectively, these data suggest that instead of resident CD11b⁺F4/80high Kupffer cells, CD11b⁺F4/80low macrophages are responsible for NOX2-mediated ROS generation in the liver of HFD-fed mice.

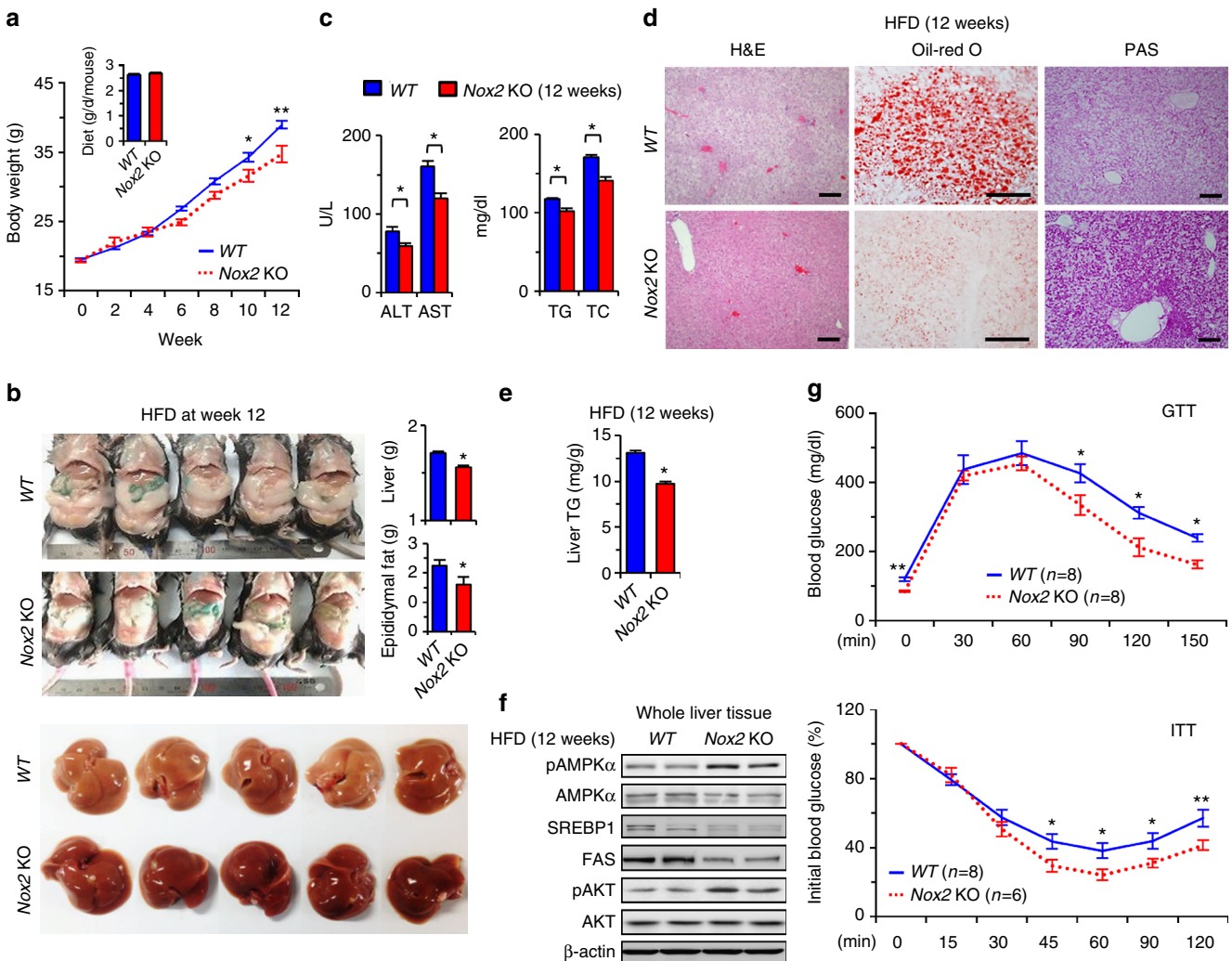

**Fig. 1** Ablation of NOX2 ameliorates high-fat diet-induced hepatic steatosis in mice. *WT* and *Nox2* KO mice were fed a high-fat diet for 12 weeks. **a** Changes in body weight and diet intake. **b** Representative gross findings and their weights of epididymal fat and liver in *WT* and *Nox2* KO mice at week 12. **c** Blood chemistry analyses for alanine aminotransferase (ALT), aspartate aminotransferase (AST), triglyceride (TG), and total cholesterol (TC). **d** Sectioned liver tissues stained with hematoxylin and eosin (H&E), oil-red O (Oil-Red O), and periodic acid-Schiff (PAS). Bar = 100 μm. **e** TG levels measured in whole liver tissues. **f** Liver tissues subjected to Western blotting. **g** Glucose tolerance tests (GTTs) and insulin tolerance tests (ITTs) performed after 16 h of fasting. Data are representative of three independent experiments using 5 (**a**–**f**) or 6–8 (**g**) mice per group. Data are expressed as the mean ± s.e.m. and analyzed by Student's *t*-test, *$P < 0.05$, **$P < 0.01$ in comparison with the corresponding controls

**Pro-inflammatory phenotype of CD11b⁺F4/80^low macrophages**. To validate the notion that CD11b⁺F4/80^low macrophages rather than CD11b⁺F4/80^high Kupffer cells are responsible for NOX2-dependent ROS generation in response to HFD-mediated liver injury, macrophage polarization markers were analyzed in hepatic macrophages isolated from *WT* mice (C57BL/6), using fluorescence-activated cell sorting (FACS). As previously reported[19], two types of hepatic macrophages were identified as CD11b⁺F4/80^high (or CD11b⁺Ly6G⁺Ly6C^low) Kupffer cells and CD11b⁺F4/80^low (or CD11b⁺Ly6G⁻Ly6C^high) macrophages (Fig. 3a). CD11b⁺F4/80^high Kupffer cells showed higher expression of CD206 than CD11b⁺F4/80^low macrophages, with no expression detectable in hepatic lymphocytes, as a reflection of a more pro-inflammatory phenotype in infiltrating macrophages than in Kupffer cells (Fig. 3a). Next, we assessed morphological differences between Kupffer cells and macrophages by sorting and staining with Giemsa solution. Under the microscope, CD11b⁺F4/80^low macrophages were identified with their small cytoplasm and indented nuclei, whereas CD11b⁺F4/80^high Kupffer cells showed abundant cytoplasm and marginal location of ovoid

nuclei (Fig. 3b; Supplementary Fig. 2a, b). In qRT–PCR analyses, CD11b⁺F4/80^low macrophages displayed higher expression of *Nox2*, *Tlr4* and *Cd14* than CD11b⁺F4/80^high Kupffer cells (Fig. 3c). In addition, gene expression of pro-inflammatory markers such as *Tnf-α* and *Il-1β* was higher, whereas expression of *Il-10* was significantly lower in CD11b⁺F4/80^low macrophages than in CD11b⁺F4/80^high Kupffer cells (Fig. 3d). Interestingly, similar findings were still detected in freshly isolated hepatic macrophages and Kupffer cells isolated from mice fed a HFD for 12 weeks (Supplementary Fig. 2c). Histogram data showed that expression of TLR4 was similar but expression of CD14 and CD11c was higher in macrophages than those of Kupffer cells (Fig. 3e). All these data indicate that CD11b⁺F4/80^low macrophages show more inflammatory phenotype than CD11b⁺F4/80^high Kupffer cells in liver of HFD-fed *WT* mice.

**Endocytosis of TLR4 and NOX2 in CD11b⁺F4/80^low cells by palmitate**. To explore the roles of TLR4 and NOX2 in ROS generation, isolated macrophages and Kupffer cells of *WT*, *Tlr4* KO and *Nox2* KO mice were treated with palmitate and then ROS generation

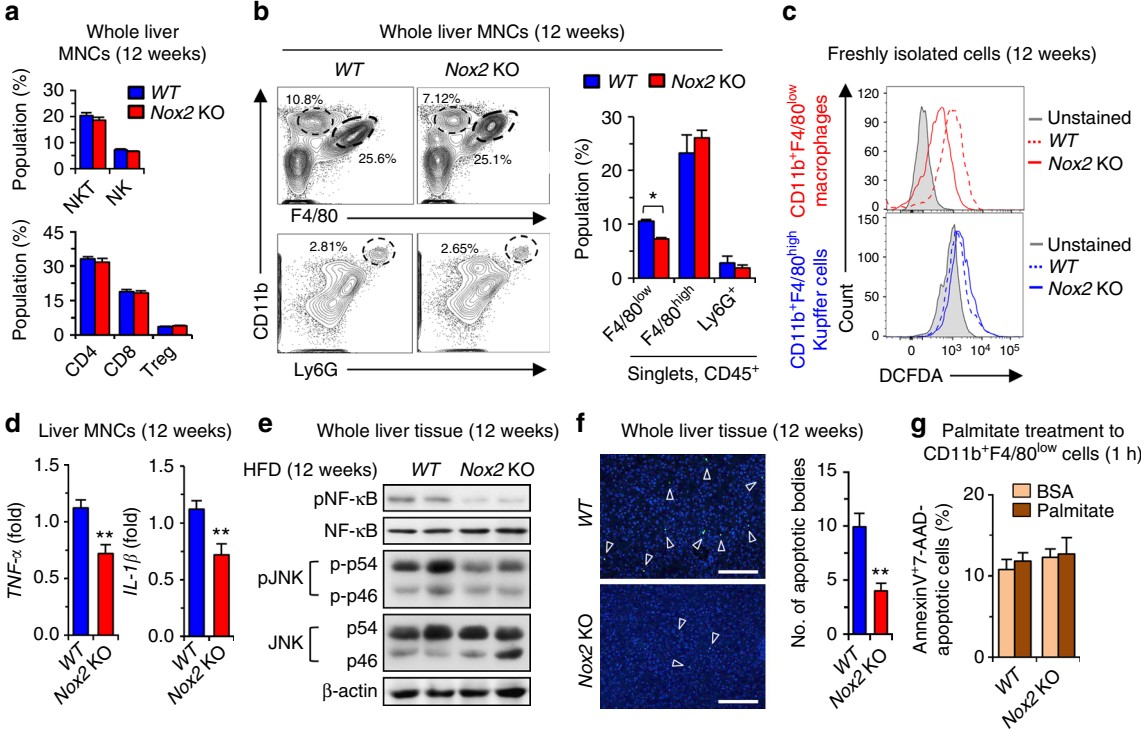

**Fig. 2** NOX2 deficiency decreases inflammatory response in CD11b⁺F4/80^low macrophages in mice fed a high-fat diet. *WT* and *Nox2* KO mice were fed a high-fat diet for 12 weeks. **a**, **b** Isolated whole liver MNCs were subjected to flow cytometry analyses. **c** The generation of ROS was monitored by DCF fluorescence in freshly isolated CD11b⁺F4/80^high Kupffer cells and CD11b⁺F4/80^low macrophages. **d**, **e** Whole liver MNCs and tissues were subjected to qRT–PCR and Western blotting, respectively. **f** Apoptotic bodies were assessed and counted after TUNEL staining (average number of 5 fields under ^χ200 magnification). Bar = 200 μm. **g** After treatment with palmitate for 1 h, CD11b⁺F4/80^low macrophages stained with Annexin V and 7-AAD were analyzed by flow cytometry. Data are representative of three independent experiments using 5 (**a–f**) and 3 (**g**) mice per group. Data are expressed as the mean ± s. e.m. and analyzed by Student's *t*-test, **$P < 0.01$ in comparison with the corresponding controls

was assessed. As expected, CD11b⁺F4/80^low macrophages of *WT* mice produced ROS, while CD11b⁺F4/80^low macrophages of *Nox2* KO and *Tlr4* KO mice did not (Fig. 4a; Supplementary Fig. 3a). More interestingly, CD11b⁺F4/80^high Kupffer cells from any of the three strains (*WT*, *Tlr4* KO and *Nox2* KO mice) failed to produce ROS in response to palmitate treatment (Fig. 4a). In qRT–PCR analyses, *dynamin* expression was significantly increased in CD11b⁺F4/80^low macrophages compared with CD11b⁺F4/80^high Kupffer cells in *WT* mice (Fig. 4b). Blocking dynamin by dynasore resulted in a significant reduction of ROS generation in CD11b⁺F4/80^low macrophages of *WT* mice (Fig. 4c).

Next, we analyzed RAW 264.7 mouse macrophages because of their similarity to CD11b⁺F4/80^low macrophages (Fig. 4d). Palmitate treatment of RAW 264.7 cells increased ROS generation and *Il-1β* expression and these effects were abrogated by dynasore treatment (Fig. 4e). Immunostaining revealed the wide distribution but discrete localization of TLR4 and NOX2 proteins in the cytoplasm of control RAW 264.7 macrophages (Fig. 4f, upper lane). Palmitate treatment resulted in the internalization of these proteins to nucleus and also their increased co-localization in the perinuclear areas (Fig. 4f, middle lane). Dynasore treatment inhibited the internalization and co-localization of these proteins (Fig. 4f, lower lane; Supplementary Fig. 3b). These data suggest that dynamin-dependent endocytosis of both NOX2 and TLR4 proteins is important for ROS generation in macrophages, but not Kupffer cells, in response to palmitate treatment.

**Palmitate-mediated ROS is produced in a TLR4/NOX2-dependent manner.** Immunoblots of RAW 264.7 cells revealed

palmitate-induced significant internalization of TLR4 and NOX2 proteins from membrane fraction to cytosol, but a direct interaction between TLR4 and NOX2 proteins was not observed (Fig. 5a). In addition, phosphorylation of NF-κB and JNK (pNF-κB and pJNK) and expression of *Nlrp3* and *Il-1β* mRNAs were increased by palmitate treatment, but these effects were absent in *Nox2*- or *Tlr4*-depleted RAW 264.7 cells (Fig. 5a; Supplementary Fig. 4a–d). In parallel, palmitate treatment induced not only caspase-1 activation but also increased expression of *Il-1β* and *Nlrp3* in WT bone marrow-derived macrophages (BMDMs), but not in *Nox2* KO BMDMs (Fig. 5b, c). Next, we investigated molecular mechanisms of TLR4/NOX2-mediated ROS generation. In response to palmitate, siRNA-mediated depletion of *Tlr4*, *Md2*, *Nox2*, *Rac1*, and *Rac2* abolished ROS production, whereas treatment with siRNA against *Cd14*, *Myd88*, and *Trif* or the inhibition of *Cd36*, fatty acid translocase, did not affect ROS production (Fig. 5d; Supplementary Fig. 4e).

To further test whether palmitate induces MyD88/TRIF-independent, TLR4/NOX2-dependent ROS generation in vivo, we perfused the liver of *WT*, *Tlr4* KO, *Nox2* KO and *Myd88/Trif* double KO mice via the portal vein with BODIPY-labeled fatty acid analogs (C16-BODIPY) or palmitate. During the last minute of perfusion, superior and inferior vena cava were clipped so that the liver remained exposed to the perfusate, followed by rapid isolation of hepatic immune cells (Fig. 5e). We verified that C16-BODIPY was successfully delivered to all hepatic immune cells with high *Nox2* expression, including macrophages, Kupffer cells and neutrophils (Fig. 5f, lower panels; Supplementary Fig. 4f). However, only CD11b⁺F4/80^low macrophages generated ROS in response to palmitate infusion (Fig. 5f, upper panels).

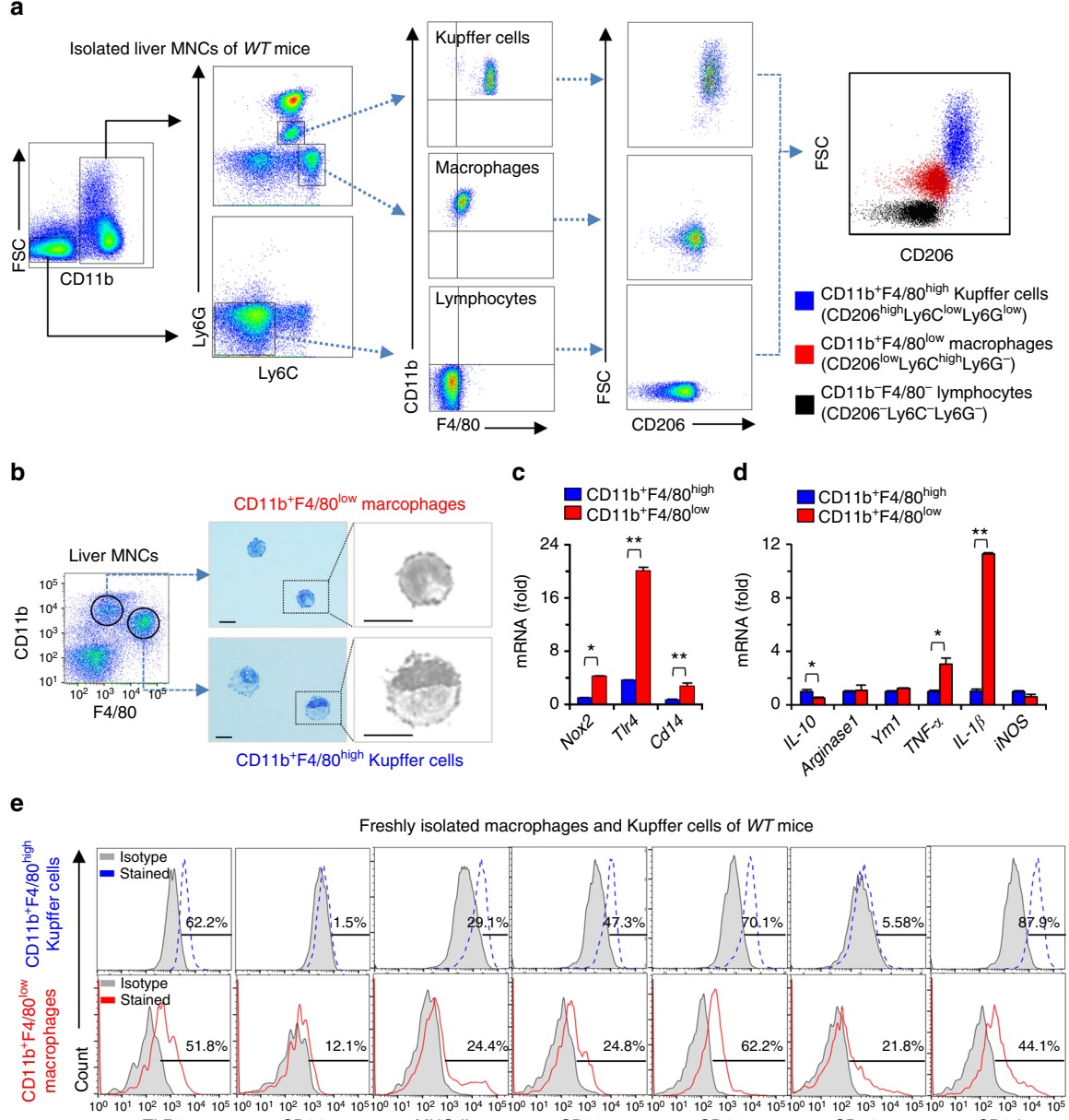

**Fig. 3** CD11b$^+$F4/80$^{low}$ macrophages present a more pro-inflammatory phenotype than CD11b$^+$F4/80$^{high}$ Kupffer cells. **a** Flow cytometry analysis of freshly isolated liver mononuclear cells (MNCs) stained for monocyte lineage markers Ly6C, F4/80, CD11b and CD206 from control *WT* mice. Prior to CD11b$^+$ gating, singlet and live cell gating were performed. **b** Isolated cells were visualized by Giemsa staining. Bar = 10 μm. **c** The gene expression of *Nox2*, *Tlr4* and *Cd14* in Kupffer cells and macrophages. **d** Relative expression of pro-inflammatory genes in Kupffer cells and macrophages. **e** Flow cytometry analysis of CD11b$^+$F4/80$^{high}$ Kupffer cells and CD11b$^+$F4/80$^{low}$ macrophages. Percentage of surface marker expression is depicted above the histograms. Data are representative of three independent experiments in vitro using isolated liver immune cells from 3 mice per group. Data are expressed as the mean ± s.e.m. and analyzed by Student's *t*-test, *$P < 0.05$, **$P < 0.01$ in comparison with the corresponding controls

Interestingly, CD11b$^+$F4/80$^{low}$ macrophages from *WT* and *Myd88/Trif* double KO mice produced similar levels of ROS in response to palmitate, whereas no such effect was observed in *Tlr4* KO or *Nox2* KO mice (Fig. 5g). All this suggests that palmitate treatment generates ROS by a MyD88/TRIF-independent, TLR4/NOX2-dependent manner.

**Palmitate binding to the monomeric TLR4–MD2 complex in macrophages**. To investigate direct binding of palmitate to the TLR4–MD2 complex, purified recombinant TLR4–MD2, MD2, and CD14 from Hi5 insect cells were incubated

with C16-BODIPY as previously reported[20, 21] and were subjected to native polyacrylamide gradient gel electrophoresis (Native PAGE). Strong fluorescence signal was observed in C16-BODIPY plus TLR4–MD2 complex, which was co-localized with the position of TLR4–MD2 complex as visualized by Commassie stain, indicating that C16-BODIPY can bind to TLR4–MD2 but not to CD14 and MD2 (Supplementary Fig. 5).

Based on the above findings, we further investigated whether palmitate binding can induce the dimerization of TLR4–MD2 complex, similar to LPS. In size-exclusion chromatography, a representative peak of monomeric human TLR4–MD2

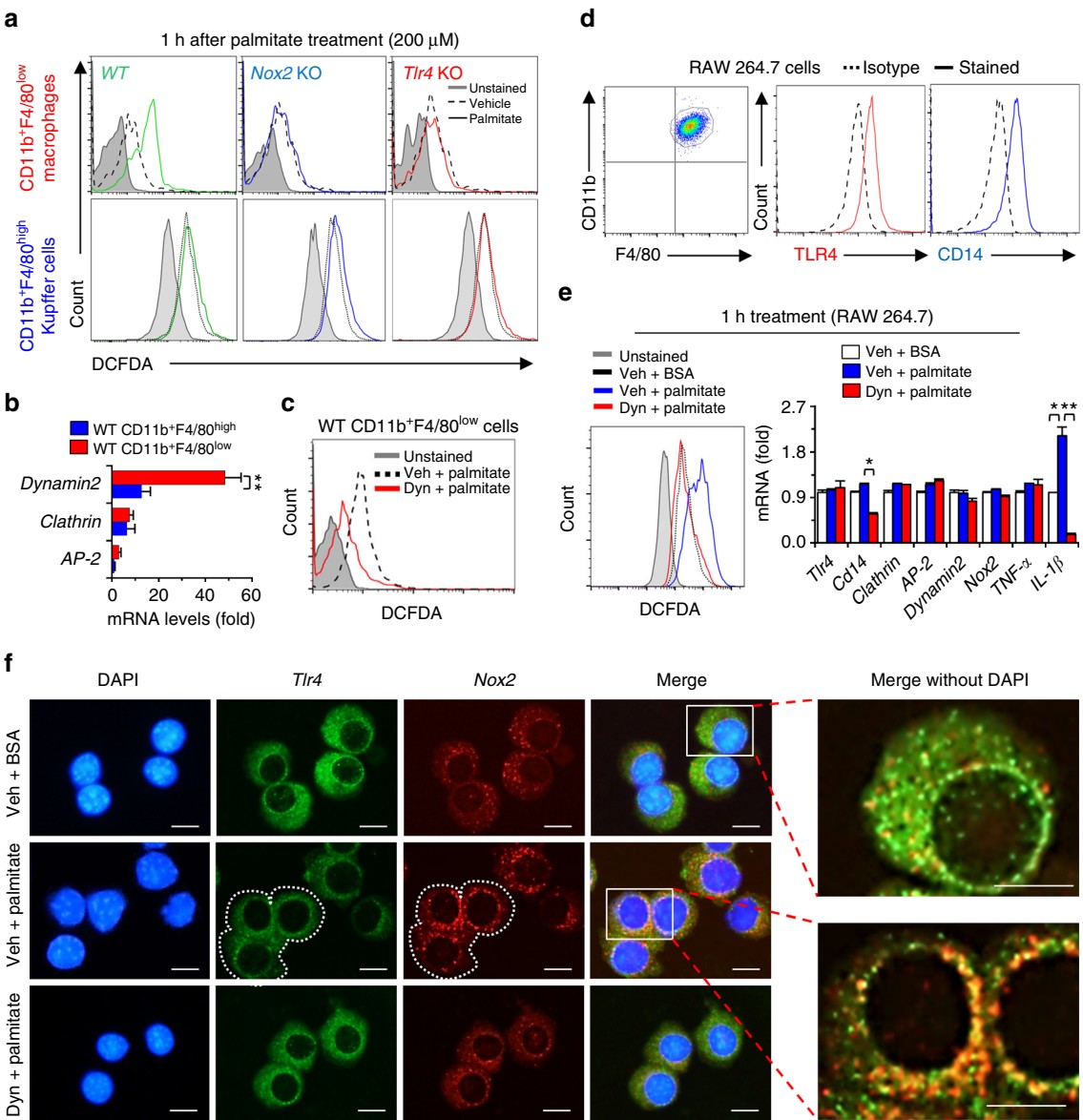

**Fig. 4** Palmitate treatment increases ROS generation in CD11b+F4/80low macrophages in TLR4 and NOX2-dependent manners. **a** Palmitate-mediated generation of reactive oxygen species (ROS) was monitored by DCF fluorescence in freshly isolated CD11b+F4/80high Kupffer cells and CD11b+F4/80low macrophages of *WT*, *Nox2* knockout (KO) and *Tlr4* KO mice. **b** Freshly isolated macrophages and Kupffer cells of *WT* mice were subjected to quantitative real-time PCR (qRT–PCR) analyses. **c** Isolated macrophages of *WT* mice were treated with palmitate with or without dynasore (Dyn) treatment. **d** RAW 264.7 macrophages were subjected to flow cytometry analyses. **e, f** RAW 264.7 macrophages were treated with palmitate (200 μM) ± dynasore (80 μM) for 1 h. Then, these cells were subjected to ROS generation assays, qRT–PCR analyses and immunostaining with antibodies of TLR4 and NOX2. Bar = 10 μm. Dotted white line indicates cell boundaries. Solid white rectangles are magnified. Data are representative of three independent experiments in vitro using isolated liver immune cells from 3 (**a–c**) mice per group. Data are expressed as the mean ± s.e.m. and analyzed by Student's *t*-test or one-way analysis of variance, *P < 0.05, **P < 0.01 in comparison with the corresponding controls

(hTLR4–MD2) complex was eluted around the 14 ml fraction compared with controls such as thyroglobulin, γ-globulin, ovalbumin, myoglobin, and vitamin B12, indicating the formation of a stable monomeric TLR4–MD2 complex (~90 kDa, 1:1) (Fig. 6a). After incubation with LPS Ra, a significantly forward-shifted peak at the 12 ml fraction was identified as dimerized hTLR4–MD2 complex, whereas the unbound monomeric hTLR4–MD2 complex was eluted at the 14 ml fraction (Fig. 6b). Surprisingly, the hTLR4–MD2 complex incubated with C16-BODIPY remained as a monomeric complex (eluted at 14 ml), despite its tight binding to C16-BODIPY, which was confirmed by fluorescence signal and Commassie staining on Native PAGE (Fig. 6c). To exclude unexpected binding of C16-BODIPY, the

hTLR4–MD2 complex was further incubated with palmitate conjugated with bovine serum albumin (BSA) or BSA only. Consistent with C-16 BODIPY, incubation with palmitate-BSA or BSA did not induce dimerization of the hTLR4–MD2 complex (Fig. 6d, e). These data suggest that unlike LPS binding to a dimerized TLR4–MD2 complex, palmitate binds to a monomeric TLR4–MD2 complex and may initiate the subsequent internalization of the TLR4–MD2 complex along with NOX2 for ROS generation (Fig. 6f).

**Endocytosis of the TLR4–MD2 complex generates ROS in human macrophages.** Similar to mouse liver, human liver tissue

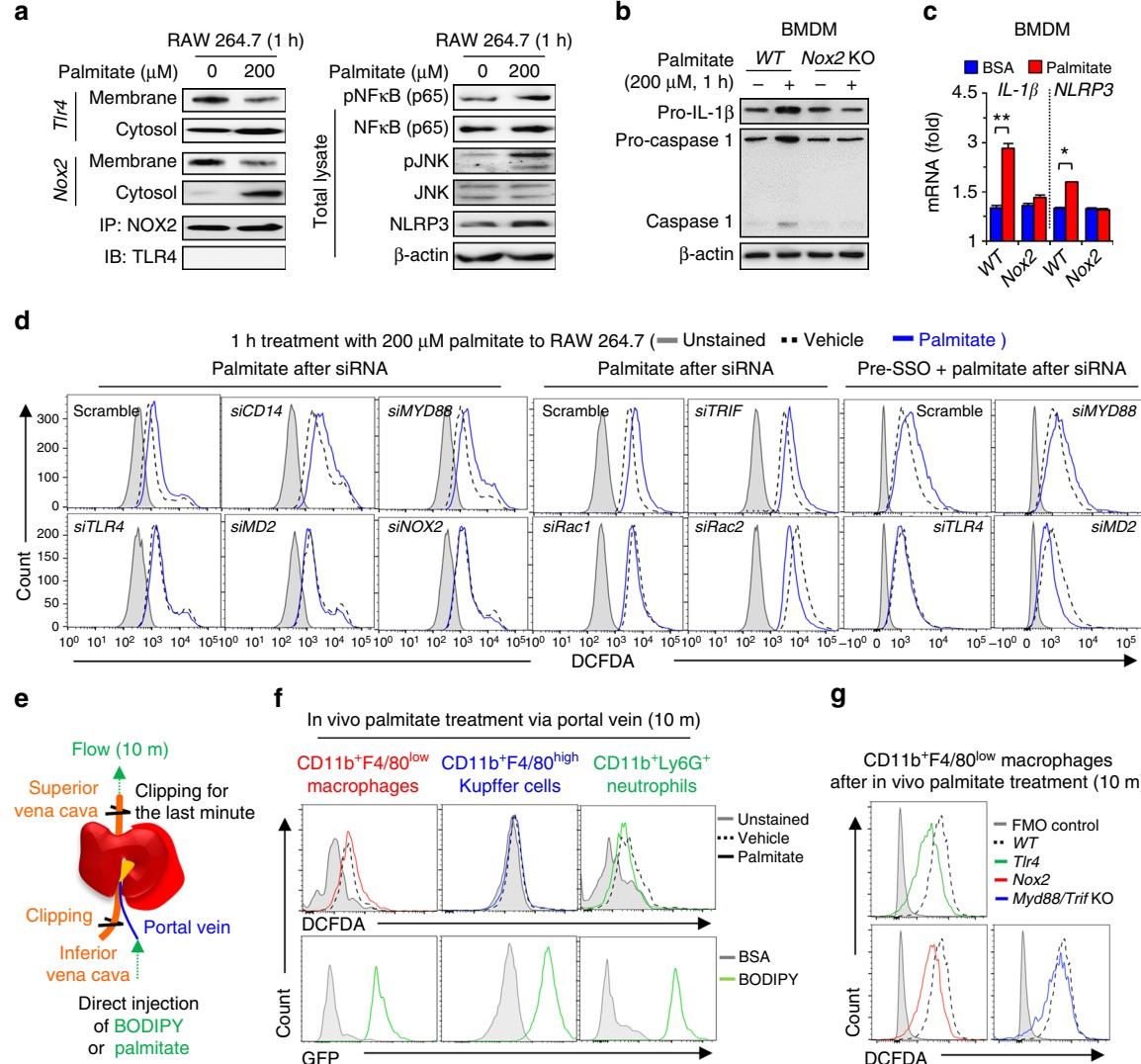

**Fig. 5** Endocytosis of palmitate/TLR4–MD2 complex generates NOX2-mediated ROS. **a** RAW 264.7 cells were stimulated with 200 μM palmitate for 1 h. Total cell lysates, membrane fractions or cytosol fractions of RAW 264.7 cells were subjected to Western blotting or immunoprecipitation (IP). With anti-NOX2, immunocomplexes from cytosolic fractions were followed by immunoblot analysis with antibody of TLR4. **b**, **c** BMDMs of *WT* and *Nox2* KO mice were stimulated with 200 μM palmitate for 1 h and then they were subjected to Western blotting and qRT–PCR analyses, respectively. **d** RAW 264.7 macrophages were subjected to FACS analyses to measure ROS generation after each siRNA silencing. To inhibit CD36-mediated palmitate uptake, RAW 264.7 cells were pre-treated with sulfosuccinimidyl oleate (SSO) for 10 min. **e** Direct perfusion of the liver with palmitate. C-16 BODIPY (10 μM) and palmitate (500 μM) were infused via the portal vein for 10 minutes. Both the superior and inferior vena cava were clipped for the last one minute to trap the solution in the liver. **f**, **g** After perfusing C16-BODIPY or palmitate through the liver of *WT*, *Tlr4* KO, *Nox2* KO and *Myd88/Trif* double KO mice, successful delivery of C16-BODIPY and ROS generation were assessed in freshly isolated CD11b⁺F4/80^low macrophages, CD11b⁺F4/80^high Kupffer cells and CD11b⁺Ly6G^high neutrophils, respectively. Data are representative of three independent experiments using isolated liver immune cells from 3 (**f**, **g**) mice per group. Data are expressed as the mean ± s.e.m. and analyzed by Student's *t*-test, *$P < 0.05$, **$P < 0.01$ in comparison with the corresponding controls

from a patient with HBV-related hepatocellular carcinoma (HCC) also had two types of macrophages such as CD68^low CD14^high macrophage and CD68^high CD14^low resident Kupffer cells, whereas only CD68^low CD14^high monocytes were found in peripheral blood mononuclear cells (PBMCs) from healthy human donor blood (Fig. 7a). Human CD68^low CD14^high macrophages showed strong ROS production compared with human Kupffer cells (Fig. 7a). Also, CD68^low CD14^high macrophages but not in CD68^high CD14^low Kupffer cells from normal hepatic tissue sample of the patient with cryptogenic HCC generated ROS in response to palmitate (Supplementary Fig. 6a, b). In addition to combining fatty acid ligand with mTLR4–MD2, mMD2, and hTLR4–MD2 (Fig. 5e; Fig. 6c), hMD2 proteins were co-incubated with C16-BODIPY and the samples were separated

by Native PAGE (Fig. 7b). Similarly, hMD2 showed a slight interaction with C16-BODIPY (Fig. 7b; Supplementary Fig. 6c, d). In human CD68^low CD14^high monocytes isolated from PBMCs, palmitate-induced ROS generation and increased expression of *Il-1β* and *Nlrp3*, and these effects were suppressed by dynasore treatment (Fig. 7c). Next, we compared hepatic macrophage population and gene expression of whole liver MNCs isolated from non-tumor/non-steatotic lesions of the patient with cryptogenic HCC and fatty liver lesions of the patient without HCC (Fig. 7d). FACS analyses revealed the same populations of CD68^low CD14^high macrophage and CD68^high CD14^low Kupffer cells with different frequencies in the two livers, but qRT–PCR analyses revealed significantly increased expression of *Tlr4*, *dynamin2*, *Rac1*, *Rac2*, *Nlrp3*, *Il-1β* and *Tnf-α* in MNCs from fatty

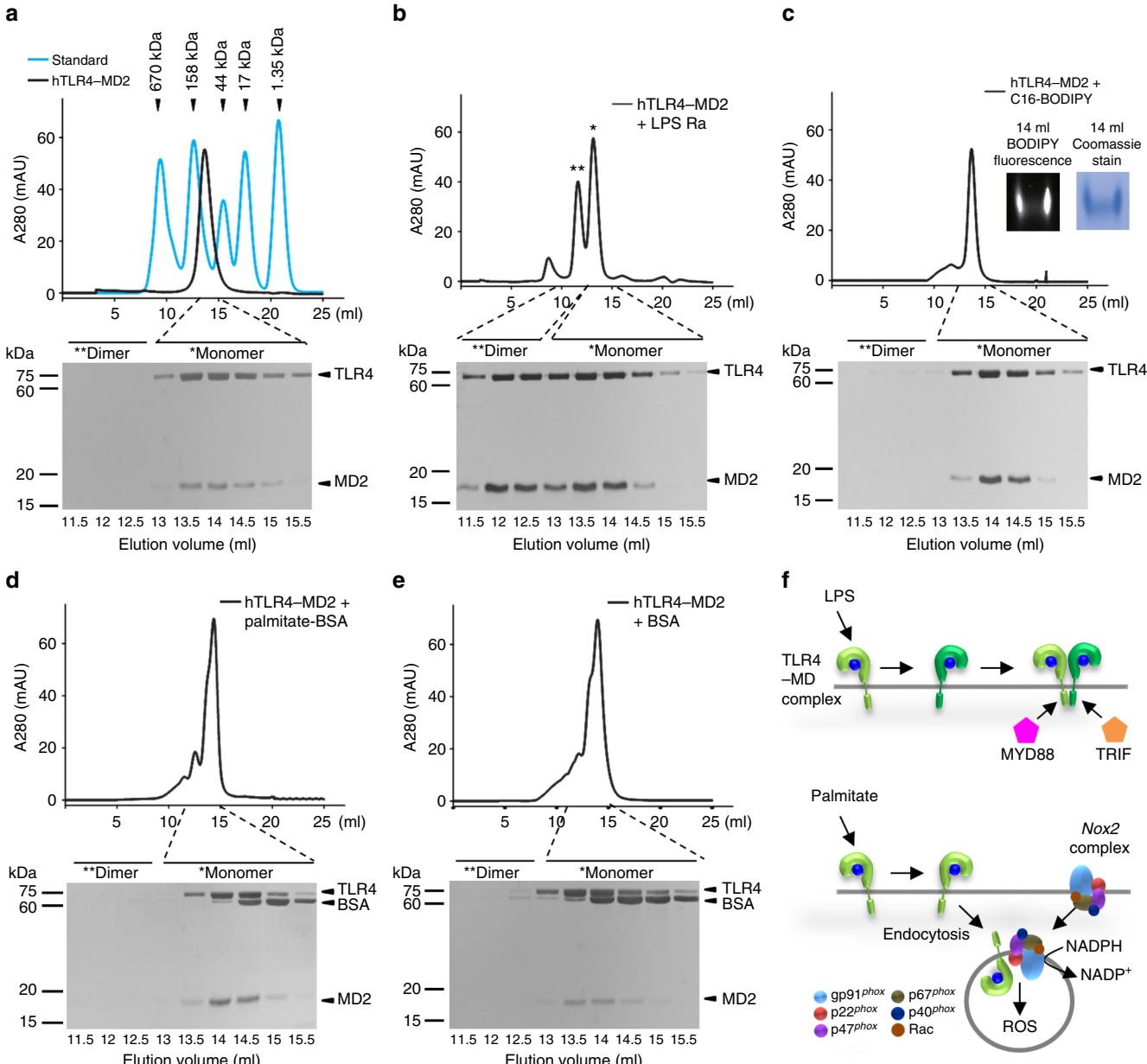

**Fig. 6** Monomeric interaction of hTLR4–MD2 complex in response to palmitate treatment. Dimerization of human TLR4-MD2 (hTLR4–MD2) complex mediated by different lipid ligands was examined by size-exclusion chromatography (Superdex 200) and SDS–PAGE with silver staining. **a** Arrowheads in chromatogram indicate each elution volume of standard proteins (thyroglobulin, 670 kDa; γ-globulin, 158 kDa; ovalbumin, 44 kDa; myoglobin, 17 kDa; vitamin B12, 1.35 kDa). The monomeric hTLR4–MD2 complexes (~ 90 kDa) were eluted at ~14 ml fraction. **b** Incubation of hTLR4–MD2 complex with LPS Ra induced the dimerization of hTLR4–MD2 complex. The dimeric hTLR4–MD2 complex by LPS was eluted at 12 ml and the remaining monomeric TLR4–MD2 complexes were eluted at 14 ml. **c-e** Incubation of hTLR4–MD2 complex with C16-BODIPY (**c**), palmitate-BSA (**d**), or BSA (**e**) could not induce the dimerization of hTLR4–MD2 complexes. Native PAGE followed by illumination at 488 nm and Coomassie staining confirmed the binding of C16-BODIPY to hTLR4–MD2 complex (**c**, inset). **f** Schematic diagram of dimeric or monomeric states of TLR4–MD2 complex in response to treatments of LPS and palmitate. The dimeric hTLR4–MD2 complex by LPS requires binding with MyD88 or TRIF, while monomeric TLR4–MD2 complex by palmitate generates ROS by NOX2 complex. Data are representative of three independent experiments

liver compared to MNCs from non-steatotic normal lesion of HCC (Fig. 7d). Moreover, in a data set downloaded from the GEO database (accession code GSE63067), gene expression of *Tlr4*, *Nox2*, *Rac1* and *Rac2* was remarkably increased in patients with non-alcoholic steatohepatitis compared with heathy controls (Supplementary Fig. 7). These data suggested that endocytosis of TLR4–MD2 complex in human CD68^lowCD14^high macrophages by palmitate-mediated ROS generation, which might be related with pathogenesis of NAFLD in human patients.

**NOX2-deficient BM transplantation attenuates hepatic steatosis.** In order to exclude off-target effects of NOX2 and to confirm the importance of NOX2 in macrophages, we generated chimeric mice by reciprocal bone marrow transplantation (BMT) using WT and *Nox2* KO mice (WT^WT, WT^NOX2, NOX2^WT, or NOX2^NOX2) as previously reported (Supplementary Fig. 8a)[22]. Body, liver and epididymal fat pad weights were significantly lower in mice with *Nox2* KO BMT (WT^NOX2 and NOX2^NOX2) than in mice with WT BMT (WT^WT and NOX2^WT)

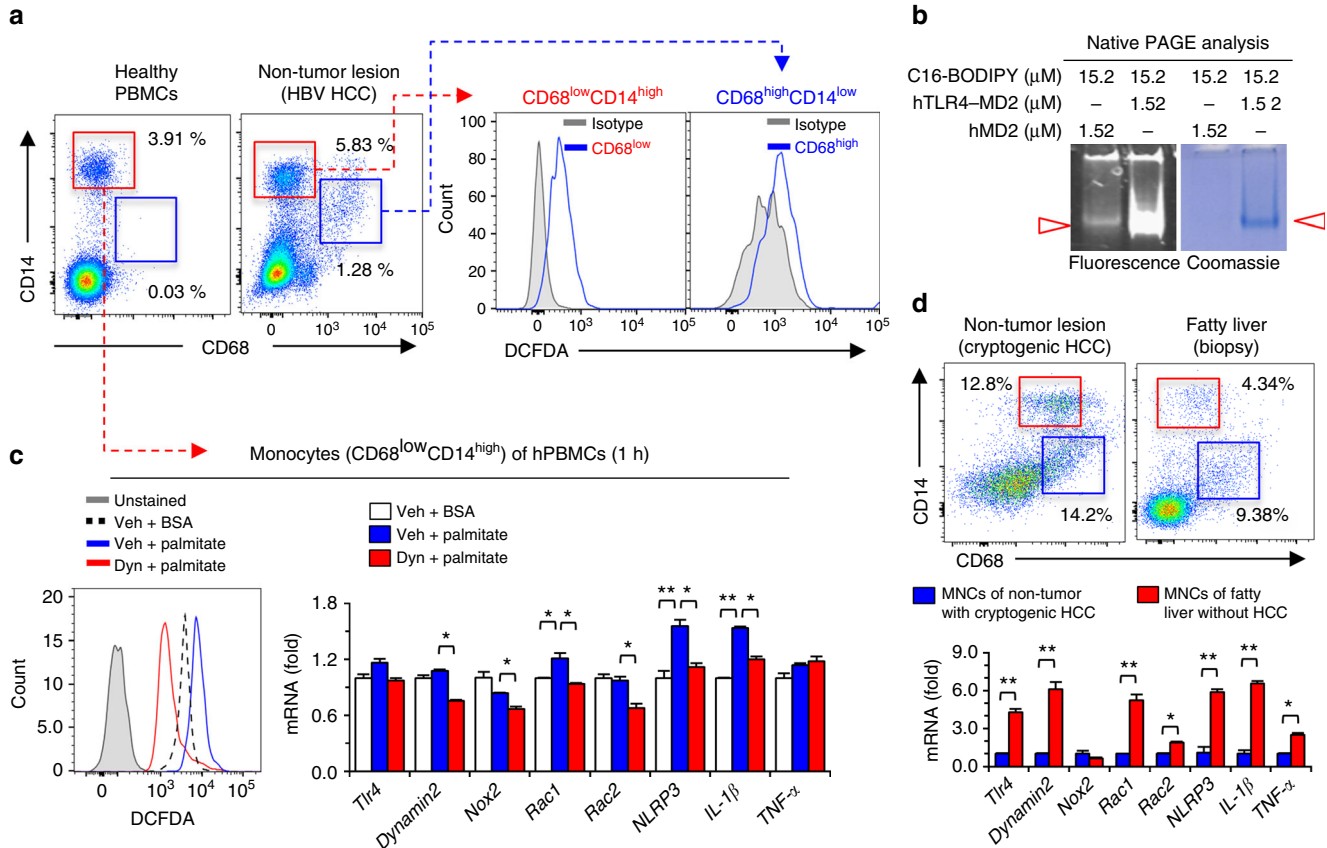

**Fig. 7** Palmitate treatment increases ROS generation in human CD68^lowCD14^high monocytes via endocytosis of palmitate/TLR4 complex. **a** Healthy human peripheral blood mononuclear cells (PBMCs) and isolated liver mononuclear cells (MNCs) from non-tumor lesions of HCC (HBV origin) were subjected to flow cytometry and assessment of ROS generation, respectively. **b** After the reaction between each protein and BODIPY-labeled fluorescent fatty acid analog (C16-BODIPY) under the indicated conditions, the samples loaded at Native gradient PAGE (4–15%) were visualized with illumination (488 nm) and Coomassie staining. **c** Healthy human PBMCs were treated with palmitate ± dynasore. Then, these cells were subjected to assessment of ROS generation and qRT–PCR analyses. **d** Freshly isolated liver MNCs from non-tumor liver lesions of primary HCC and biopsy lesions of fatty liver were subjected to flow cytometry and qRT–PCR analyses. Data are representative of three independent experiments using PBMC of healthy controls ($n = 5$) and liver MNCs of HBV ($n = 3$), cryptogenic HCC ($n = 1$) and fatty liver ($n = 1$) patients **a**, **c**, **d**. Data are expressed as the mean ± s.e.m. and analyzed by Student's $t$-test or one-way analysis of variance, *$P < 0.05$, **$P < 0.01$ in comparison with the corresponding controls

(Supplementary Fig. 8b, c). $WT^{NOX2}$ and $NOX2^{NOX2}$ mice had improved glucose tolerance as assessed by GTT (Fig. 8a) and lower serum ALT and TC, as compared to $WT^{NOX2}$ and $NOX2^{NOX2}$ mice (Supplementary Fig. 8d). The livers of $WT^{NOX2}$ and $NOX2^{NOX2}$ mice were less yellowish in color and had lower TG content than $WT^{WT}$ and $NOX2^{WT}$ mice (Fig. 8b). Coincidently, fat accumulation was remarkably reduced, whereas deposition of glycogen was greatly increased in liver sections of $WT^{NOX2}$ and $NOX2^{NOX2}$ mice (Fig. 8c). Although there were no differences in frequencies of hepatic macrophages among the groups, $Il$-$1\beta$ and $Tnf$-$\alpha$ gene expression was considerably decreased whereas $Il$-$10$ expression was increased in liver MNCs of $WT^{NOX2}$ and $NOX2^{NOX2}$ mice compared with those of $WT^{WT}$ and $NOX2^{WT}$ mice (Fig. 8d; Supplementary Fig. 8e, f). Immunoblots of phosphoproteins revealed increased phosphorylation of AMPK and AKT but decreased phosphorylation of NF-κB and JNK in liver tissues of $WT^{NOX2}$ and $NOX2^{NOX2}$ mice compared with those of $WT^{WT}$ and $NOX2^{WT}$ mice (Fig. 8e). In addition, expression of SREBP1 and FAS was significantly down-regulated in $WT^{NOX2}$ and $NOX2^{NOX2}$ mice (Fig. 8e).

## Discussion
The role of NOX2 on innate immune responses has been well described in mice and humans, yet the mechanism of palmitate-mediated ROS generation via NOX2 is not well understood, especially in transmigrating macrophages in the steatotic and insulin resistant liver. In the present study, we have demonstrated that palmitate binds a monomeric TLR4–MD2 complex in CD11b+F4/80^high and CD68^lowCD14^high macrophages of mice and humans, respectively, which leads to NOX2-mediated ROS generation by dynamin-mediated endocytosis of the TLR4–MD2 complex. Another striking finding was that the HFD or palmitate-induced ROS generation and pro-inflammatory changes selectively appeared in transmigrating liver macrophages and not in resident Kupffer cells. This, in turn, is responsible for the obesity-related steatohepatitis and insulin resistance, as summarized in Fig. 8f.

Increased inflammation is a key feature of metabolic disorders such as hepatic steatosis and insulin resistance. Recent studies have indicated that genetic depletion of NOX2 resulted in reduced visceral adiposity, diminished visceral fat macrophage infiltration and improved glucose regulation in HFD-fed male mice[23], and that a deficiency in the cytosolic NOX2 system component p47phox resulted in reduced hepatic TG in HFD-fed male and female mice[24]. In addition, Ezetimibe-mediated reduction of NOX2 activation attenuated hepatic steatosis and macrophage infiltration of db/db mice[25]. These findings suggest that NOX2 plays an important role in diet-induced obesity, hepatic steatosis and insulin signaling. In contrast, singly housed

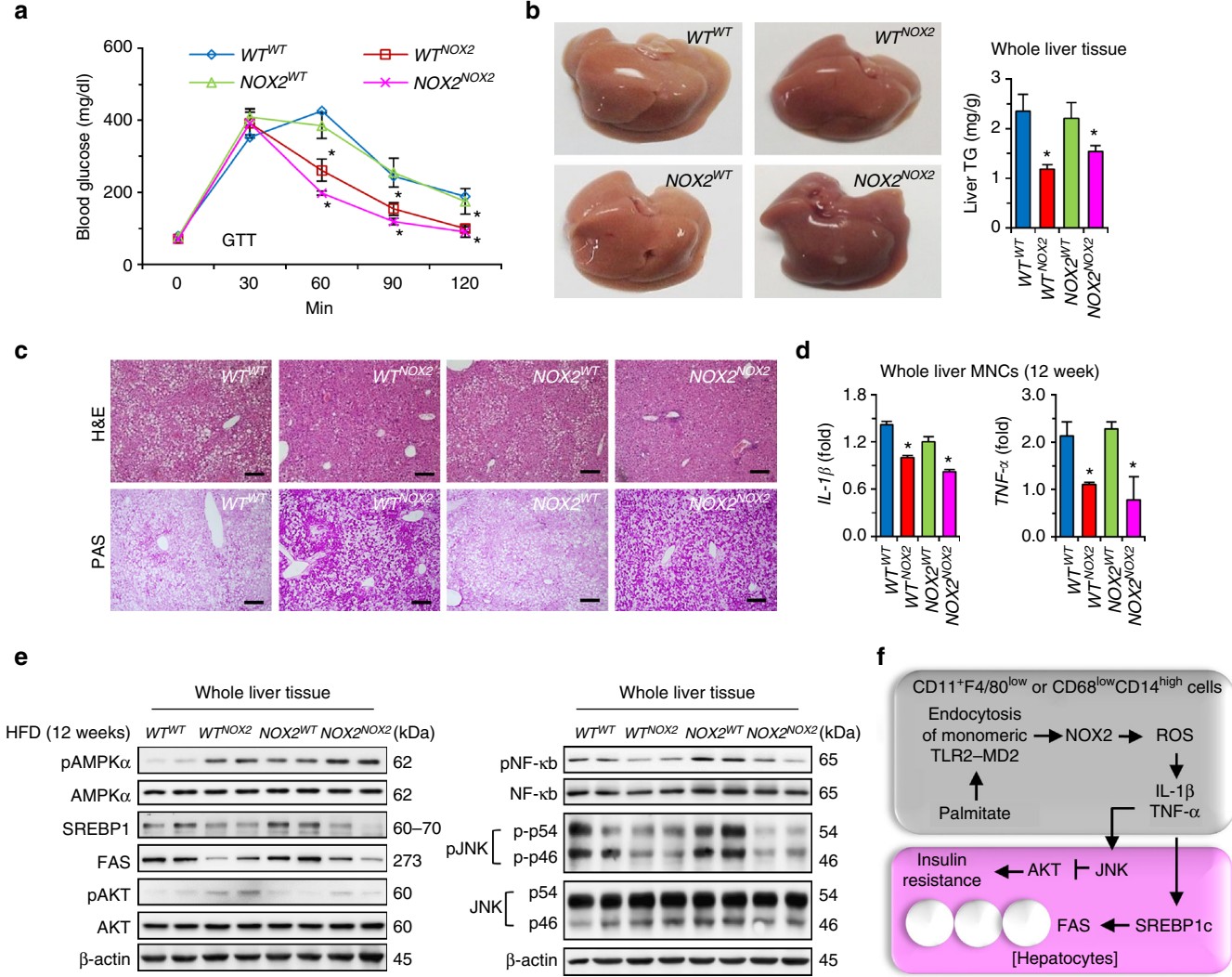

**Fig. 8** NOX2-deficient bone marrow transplantation attenuates high-fat diet-induced hepatic steatosis in mice. After reciprocal bone marrow transplantation between *WT* and *Nox2* KO mice, mice were fed a high-fat diet for 12 weeks. **a** GTT was performed after 16 h fasting. **b** Liver gross findings and hepatic contents of TG were assessed. **c** Liver sections were stained with H&E (upper panels) and PAS (lower panels). Bar = 100 μm. **d** Isolated liver MNCs were subjected to qRT–PCR analyses. **e** Whole liver tissues were subjected to Western blottings. **f** Schematic diagram of NOX2-mediated ROS generation through endocytosis of monomeric TLR4–MD2 complex in non-resident macrophages by palmitate and its involvement in the pathogenesis of hepatic steatosis and insulin resistance. Data are representative of two independent experiments using 5 mice per group. Data are expressed as the mean ± s.e.m. and analyzed by one-way analysis of variance, *P < 0.05 in comparison with the corresponding controls

*Nox2* KO mice maintained on HFD from weaning for up to 18 weeks had less epididymal white adipose tissue but more hepatic lipid accumulation, inflammatory signatures and insulin resistance than *WT* mice[26]. One should note, however, that social isolation stress in single housed mice has been shown to induce liver hypertrophy and an upregulation of lipid synthetic and downregulation of lipid oxidation pathways[27, 28], which complicates interpretation of the data in the above study[26]. Through the use of reciprocal BMT, our study clearly showed improved glucose sensing, less fat but more glycogen accumulation in hepatocytes and attenuated inflammatory responses in *WT* and *Nox2* KO mice with *Nox2* KO BMT compared with *WT* and *Nox2* KO mice with WT BMT (Fig. 8; Supplementary Fig. 8). In addition, only CD11b⁺F4/80^low macrophages generated ROS in response to in vivo perfusion with palmitate, although *Nox2* was expressed not only in these cells, but also in CD11b⁺F4/80^high Kupffer cells and CD11b⁺Ly6G⁺ neutrophils (Fig. 5f). This supports that beneficial effects of NOX2-depleted BMT on high-fat

diet-induced hepatic steatosis and insulin resistance might be due to less ROS generation of transmigrating macrophages. Using a similar BMT approach, another study demonstrated that *WT* mice with *Nox2* KO BMT had decreased ROS production in mononuclear cells and impaired neovascularization following hindlimb ischemia[29]. However, further studies using macrophage-specific conditional *Nox2* KO mice are needed to provide more definitive evidence for this hypothesis.

In addition to LPS, TLR4 can bind a broad range of ligands including saturated FFAs, which may lead to diverse signaling complexes due to the possible heterodimerization of TLR4 with other TLRs, co-receptors or accessory proteins including CD14, CD36 and MD2[14, 30, 31]. Heterodimerization between TLR4 and TLR2 or TLR4 and TLR6 has been reported in microglial cells and macrophages, in which either heterodimers of TLR4-TLR2 by hemoglobin or TLR4-TLR6 by oxidized LDL/amyloid-β triggered inflammatory responses in a MyD88-dependent manner[30, 31]. In contrast, we found that MyD88 and TRIF were not required

for palmitate-induced ROS generation in vitro and in vivo (Fig. 5d–g). Similarly, minimally oxidized LDL treatment generated NOX2-mediated ROS in peritoneal macrophages by a TLR4-dependent, MyD88-independent manner[11]. In cases of CD14-dependent endocytosis of TLR4, LPS-mediated dimerization of the TLR4–MD2 complex is essential, in which CD14, Syk and PLCγ2 play critical roles, whereas TLR4 signaling is not necessary for TLR4 endocytosis in dendritic cells and bone marrow-derived macrophages[32, 33]. Our data show that the TLR4–MD2 complex was required, whereas CD14 and CD36 were dispensable for the palmitate-induced, NOX2-mediated ROS production (Fig. 5d). Furthermore, unlike LPS, palmitate binds the momomeric TLR4–MD2 complex (Fig. 6), which is a novel type of signaling complex of palmitate–TLR4–NOX2 in macrophages.

Some reports suggested that activation of TLR2 or TLR4 signaling might be attributed to endotoxin contamination in dietary saturated fatty acids[14, 34]. However, in contrast to LPS, dimerization of the TLR4–MD2 complex was not observed in treatments with C16-BODIPY, palmitate-BSA or even BSA, which excludes the role of contamination of endotoxins (Fig. 6). Similar to the dynamin-mediated endocytic pathway of LPS-TLR4[15], internalization of TLR4–MD2 complex was also dynamin dependent in both mouse macrophages and human monocytes, as indicated by the suppression of palmitate–TLR4–NOX2-mediated ROS generation by dynasore, an inhibitor of dynamin (Fig. 4e; Fig. 7c). Endocytosis by dynamin also plays important roles in NOX2-mediated ROS generation in meningococcal endotoxin-treated neutrophils and hypoxia-stimulated endothelial cells, respectively[35, 36]. We further demonstrated that palmitate strongly binds with monomeric TLR4–MD2 complex and MD2 protein in mouse and human (Fig. 6; Supplementary Fig. 5; Fig. 7b). Our findings are in agreement with those in a recent study demonstrating direct and dose-dependent binding of palmitate with protein MD2 to activate TLR4 signaling in thioglycollate-induced peritoneal macrophages[37].

Regarding activation of NOX2 after binding TLR4 with palmitate, either *Tlr4* knockdown or *Nox2* deletion remarkably reduced palmitate-induced ROS, suggesting a reciprocal interplay between NOX2 and TLR4 (Fig. 4a). There is ample evidence that in response to diverse stimuli, clustering of lipid rafts (LRs) in the plasma membrane drives NOX assembly and aggregation, and LPS stimulates the association of TLR4 and its accessory proteins with LRs, triggering a signal cascade[38–40]. Our data indicate that the TLR4 and NOX2 proteins do not co-localized in the plasma membrane under resting conditions, but display increased co-localization, albeit not direct binding, following palmitate treatments as they translocated to the perinuclear area (Fig. 4f; Fig. 5a). This suggests that TLR4 bound by palmitate might induce clustering of LRs, which are also enriched in components of the NOX2 complex, leading to the internalization of these molecules to generate ROS. However, further studies are needed to elucidate the exact mechanism involved.

Interesting studies have suggested that in vivo and in vitro HFD feeding and treatments with palmitate prime NLRP3 inflammasome via TLR2- and TLR4-mediated NF-κB signaling pathways in macrophages, dendritic cells and Kupffer cells to upregulate expression of *Il-1β* and *Nlrp3*, leading to peripheral inflammation, obesity, hepatic steatosis and insulin resistance[41–43]. Similarly, NOX activation contributes to NLRP3 inflammasome activation. In LPS-primed human and mouse macrophages, NOX4-mediated increased expression of carnitine palmitoyltransferase 1A promotes NLRP3 inflammasome activation by fatty acid oxidation-induced mitochondrial ROS generation[17]. In addition, NOX2 is necessary for palmitate-induced NLRP3 inflammasome activation in endothelial cells[44]. These reports

suggest that NLRP3 inflammasome activation might be related to activation of TLRs and NOXs. However, all priming-mediated ROS generation and inflammatory responses in macrophages were assessed at later time points (more than 6 h) after palmitate treatment, whereas NOX2-mediated ROS production plateaued at 45 to 60 min in macrophages, and RAW 264.7 cells[35, 45]. Thus, in the present study, we assessed early changes of ROS production and gene expression in human and mouse macrophages including RAW 264.7 cells at 1 h after palmitate treatment. ROS production, activation of NF-κB and JNK, and increased expression of *Il-1β* and *Nlrp3* mRNA were detected within 1 h of palmitate treatment, and these early effects were clearly independent of MyD88 and TRIF (Fig. 5; Supplementary Fig. 4) and could be reversed by depletion of NOX2 and TLR4 or dynasore treatment (Fig. 4e; Fig. 5b, c; Supplementary Fig. 4b, c; Fig. 7c). Furthermore, in vivo intraportal perfusion with palmitate resulted in increased ROS generation only in CD11b+F4/80low macrophages in a TLR4/NOX2-dependent, MyD88/TRIF-independent manner, although NOX2 was highly expressed in macrophages, Kupffer cells and neutrophils in the liver (Fig. 5f, g). All these findings suggest that early priming of NLRP3 inflammasome by palmitate-TLR4-NOX2-mediated ROS in macrophages enhances further inflammatory stress signals including LPS-mediated canonical signaling of TLR4 at later phase or amplifies inflammatory responses of macrophages and other immune cells (e.g., Kupffer cells) by either an autocrine or paracrine manner.

According to the literature[2, 3], Kupffer cells and macrophages both support the process of inflammation and insulin resistance in liver. However, our study clearly shows that rather than CD11b+F4/80high Kupffer cells, CD11b+F4/80low macrophages are responsible for NOX2-dependent ROS production in response to HFD-mediated liver injury and palmitate treatment (Fig. 2c; Fig. 4a). In addition, CD11b+F4/80high Kupffer cells did not generate ROS in response to palmitate (Fig. 4a). Although we did not investigate the underlying mechanism, the inability of palmitate to induce dynamin expression maybe a factor. Nevertheless, our data may explain why enriched Kupffer cells in liver show resistance to elevated circulating free fatty acids after HFD feeding to mice.

In addition to liver, NOX2 complex including NOX subunit proteins (e.g. p47phox) are widely expressed in multiple organs and are involved in the pathogenesis of infectious diseases. LPS-mediated NOX2 activation has been suggested in pulmonary vascular endothelial cells and cardiac fibroblast, subsequently leading to lung inflammation and cardiac fibrosis, respectively[46, 47]. Similarly, *Nox2* deletion in muscle tissue alleviates insulin resistance by reducing oxidative stress in HFD-fed mice[48]. Moreover, *Nox2* deletion in *apolipoprotein E* KO mice reduces atherosclerotic plaque development and plasma lipids by decreasing ROS in aortic endothelial cells and macrophages of HFD-fed mice[49]. Furthermore, palmitate treatment induced dysfunction and apoptosis of pancreatic β cells via NOX2-mediated ROS[50]. On the contrary, in response to mycobacteria-induced TLR2 activation, *Nox2* depletion in macrophages prominently abrogated antimicrobial activity by decreasing production of cytokines and chemokines[51].

In summary, our findings shed new light on the pathophysiologic mechanisms of TLR4 and NOX2. We have demonstrated an interplay between palmitate and monomeric TLR4–MD2 complex in mediated ROS generation selectively in macrophages but in Kupffer cells, via dynamin-dependent endocytosis of NOX2 and TLR4, and the key role of this signaling in promoting hepatic steatosis and insulin resistance in NAFLD. Therefore, our findings imply that endocytosis of TLR4 and NOX2 could be a novel therapeutic target for NAFLD, including hepatic steatosis and insulin resistance.

## Methods

**Mice.** Male C57BL/6 based *WT, Nox2* (gp91$^{phox}$) and *Trl4* KO mice were purchased from Jackson Laboratories (Bar Harbor, ME, USA), *MyD88* KO and *Trif* KO mice on C57BL/6 background were provided by Dr. Shizuo Akira, and *MyD88/Trif* double KO mice were kindly generated by Professor Suk-Jo Kang. These mice were maintained on a regular light-dark cycle (12-hour light/12-hour dark) at 24 °C with 40–60% humidity in a specific pathogen-free animal facility at Korea Advanced Institute of Science and Technology (KAIST, Daejeon, Republic of Korea). All experimental protocols including ethics were approved by the institutional Animal Care and Use Committee (IACUC) of KAIST (No. KA2011-40). The 8-week-old male mice (5–8 mice) were fed ad libitum with normal chow diet (fat content ranges from 4 to 6%) or high-fat diet (Harlan Teklad, TD 06414, Madison, WI, USA) which approximately 60% of total calories came from fat for 12 weeks. Mice with arthritis, which occasionally occurred in NOX2 mice, were ejected from test group. Body weight and diet intake were measured twice a week during this period.

**Intraportal perfusion with palmitate.** To circulate palmitate throughout the liver, hepatic blood was flushed out by brief perfusion with a buffer solution (1× PBS with antibiotics) through the portal vein of anesthetized *WT, Tlr4* KO, *Nox2* KO and *MyD88/Trif* double KO mice. Then, 10 ml of a solution of 500 µM palmitate or 10 µM C16-BODIPY was infused for 10 min. For the last one minute, both the superior and inferior vena cava were clipped, trapping the solution in the liver. Next, the extracted liver was dissociated into single cells through Gentle MACS/ Mouse liver dissociation kit (Miltenyi Biotec, Bergisch Gladbach, Germany). Isolated cells were stained with antibodies including DCF-DA and surface antibodies. Production of ROS was measured immediately by flow cytometry (LSRII, BD Bioscience, San Jose, CA, USA).

**Bone marrow transplantation.** Chimeric mice were generated by reciprocal bone marrow transplantation. Mice were irradiated with single dose 9 Gy and then were reconstituted with bone marrow cells ($2 \times 10^6$ per mouse) of femur and tibias of donor mice through tail vein injection. After allowing to rest 8 weeks, chimerism was confirmed by isolated liver MNCs of mice (Supplementary Fig. 8a).

**Serum biochemical measurements.** Serum was collected from blood and assayed for levels of alanine aminotransferase (ALT), aspartate aminotransferase (AST), triglyceride (TG), and total cholesterol (TC) using VET Test Chemistry Analyzer according to manufacturer's instruction (IDEXX Laboratories, Westbrook, ME, USA).

**Histological analyses.** Parts of left and medial lobes were fixed with 10% neutral buffered formalin or prepared using the Tissue-Tek OCT compound (Sakura, Tokyo, Japan). After deparaffinization and rehydration, paraffin sections with 5 µm thickness were stained with hematoxylin and eosin (H&E), or Periodic Acid-Schiff (PAS) reagents (Sigma-Aldrich, St. Louis, MO, USA), whereas frozen sections with 10 µm thickness were subjected to oil-red O staining (Sigma-Aldrich, St. Louis, MO, USA) for visualization of lipid droplets.

**Liver TG levels.** Hepatic lipids were extracted from 50–100 mg of liver tissues using chloroform/methanol mixture (2/1 ratio). Lipid extract was lyophilized by nitrogen gas and then dissolved in 5% fatty acid-free BSA. Resuspended lipid was used for assessments of TG level using VET Test Chemistry Analyzer according to the manufacturer's instructions (IDEXX Laboratories, Westbrook, ME, USA).

**Glucose tolerance test and insulin tolerance test.** For the glucose or insulin tolerance test, the mice were fasted from 18:00 PM to 10:00 AM (16 h) or 8:00 AM to 12:00 PM (4 h), respectively, and allowed free access to water. Glucose (2 g kg$^{-1}$; Sigma-Aldrich St. Louis, MO, USA) or 1 U kg$^{-1}$ insulin (Humalog, Lilly, Indianapolis, IN, USA) was injected intraperitoneally to mice. Tail blood glucose levels were measured by using a glucometer (Allmedicus, Anyang, South Korea) at 0, 30, 60, 90, 120 and 150 min after glucose injection or at 0, 15, 30, 45, 60, 90, and 120 min after insulin injection.

**siRNA transfection.** Target cells were transfected with 20 nM of siRNA against *Nox2, Tlr4, Rac1, Rac2, Myd88, Trif, Md2,* and *Cd14* (Bioneer, Daejeon, Korea; Supplementary Table 1) using the Lipofectamine RNAiMAX transfection reagent (ThermoFisher Scientific, Waltham, MA, USA) according to the manufacturer's instructions. After determining knockdown efficiency by quantitative RT–PCR at 24 h-transfection, experiments were performed.

**Quantitative RT–PCR.** Total RNA was isolated from liver tissues or cells with TRIzol reagent (Thermo Fisher Scientific, Waltham, MA, USA) or RNeasy Mini kit (Qiagen, Hilden, Germany) in accordance of the manufacturer's instructions. The same quantity of total RNA was reverse-transcribed to cDNA using amfiRivert cDNA synthesis master mix (GenDEPOT, Houston, TX, USA). Quantitative RT–PCR was performed by SYBR Green Real-time PCR Master Mix (Toyobo,

Osaka, Japan). To quantify transcription, the mRNA expression levels of the target genes were normalized to β-actin or 18 S. The primers used in this study were listed in Supplementary Table 2. All samples were run in duplicate and the relative gene expression calculated as $2^{-\Delta CT}$ was expressed as fold increased over control samples.

**Western blotting.** Total protein samples were isolated from frozen liver tissue and cultured cells using RIPA lysis buffer (30 mM Tris, pH 7.5, 150 mM NaCl, 1 mM PMSF, 1 mM Na$_3$VO$_4$, 10% SDS, 10% glycerol) containing protease and phosphatase inhibitors (Roche, Basel, Switzerland). Samples were separated in a 10% SDS-polyacrylamide gel electrophoresis and transferred onto nitrocellulose membrane. Membranes were blocked in 5% milk, incubated with primary antibodies at 1:1000 in PBST. pAMPKα (#2535), AMPKα (#2532), FAS (#3180), pAKT (#9271), AKT (#9272), pNF-κBp65 (#3033), NF-κBp65 (#4764), pJNK (#9251), JNK (#9252), Caspase-1 (#2225), IL-1β (#12507), NLRP3 (#15101) and β-actin (#4970) (Cell Signaling Technology, Danvers, MA, USA), SREBP1 (#MA5-16124) (US Biological Life Science, Marblehead, MA, USA) and TLR4 (#SC-293072) (Santa Cruz Biotechnology, CA, USA). Comparative amount was normalized with β-actin. According to the manufacturer's instructions, membrane proteins were isolated by using the membrane protein isolation kit (Thermo Fisher Scientific, Waltham, MA, USA) and the samples underwent for co-immunoprecipitation assays (Co-IP). Co-IP was performed using a commercial kit Pierce™ Co-Immunoprecipitation Kit (Thermo Fisher Scientific, MA, USA) In brief, proteins cross-linked to target proteins were immune-precipitated with 10 µg anti-NOX2 antibody (#SC-130543) (Santa Cruz Biotechnology, CA, USA). After this cross-reaction, mixtures incubated with protein A/G PLUS agarose beads (#SC-2003) (Santa Cruz Biotechnology, CA, USA). The immunoprecipitated proteins were isolated by boiled sample buffer and were followed by 10% SDS-PAGE.

**Primary macrophage isolation.** To isolate infiltrating macrophages and resident Kupffer cells in liver, three different methods were used. All resuspended cells were further sorted by MACS (Miltenyi Biotec, Bergisch Gladbach, Germany), Magnisort (Affymetrix, Santa Clara, CA, USA) or FACS Aria II (BD Bioscience, San Jose, CA, USA) by using antibodies of APC-Cy7 tagged anti-mouse CD11b (clone M1/70) and PE-Cy7 conjugated anti-mouse F4/80 (Clone BM8) (BD pharmingen and affymetrix). First, for isolation of infiltrated macrophages, mouse liver mononuclear cells (MNCs) were isolated with density gradient Percoll solution (GE Healthcare Life Science, Chicago, IL, USA). In brief, mouse livers were homogenized and filtered through a 70 µm nylon cell strainer (BD Bioscience, San Jose, CA, USA). After removing debris, hepatic non-parenchymal cells were collected and suspended in 40% Percoll. The cell suspension was gently overlaid onto 70% Percoll and centrifuged at 4 °C for 25 min at 3000 r.p.m. Liver MNCs were collected from the interface. Then, liver MNCs were resuspended in fresh PBS after RBC lysis. Second, to get resident Kupffer cells, mouse livers were perfused in situ first with EGTA solution (5.4 mM KCl, 0.44 mM KH$_2$PO$_4$, 140 mM NaCl, 0.34 mM Na$_2$HPO$_4$, 0.5 mM EGTA, 25 mM Tricine, pH 7.2), followed by the perfusion collagenase buffer (Worthington Biochemical Corporation, Lakewood, NJ, USA) and the digestion buffer (0.009% collagenase type I in HBSS with 0.02% DNase I) at 37 °C for 30 min. The cell suspension was filtered through a 70 µm cell strainer and centrifuged at 400 rpm for 5 minutes at room temperature to remove hepatocytes. The supernatant was transferred to a new tube and centrifuged at 400 g for 10 minutes at 4 °C, and then the pellet was resuspended in 6 ml of 11.5% Optiprep (Sigma-Aldrich, St. Louis, MO, USA) and loaded carefully onto 6 ml of 20% Optiprep and centrifuged at 1600 g for 17 min at 4 °C. The cellular fraction in the interface between 11.5 and 20% Optiprep were gently collected. Finally, collected mouse and human liver tissues were dissociated by gentle MACS with Liver dissociation kit (Miltenyi Biotec, Bergisch Gladbach, Germany). Gathered liver tissues were washed with media (stable glutamine contained media; RPMI or low glucose DMEM), and were mixed with pre-warmed media and enzyme mixture. Then liver tissues were roughly chopped, and shook under gentle MACS dissociator according to the manufacturer's instructions.

**BMDM generation.** BMDMs were prepared. Collected bone marrow cells ($1.5 \times 10^7$) from the tibias and femurs of *WT* and *Nox2* KO mice were seeded on 100 mm culture dish and incubated for 7 days in RPMI medium containing 30% of L929 conditioned medium. BMDMs were gently harvested by scraping with a cell lifter. Collected BMDMs were resuspended in complete RPMI medium and counted for use in additional experiments.

**Measurements of reactive oxygen species by palmitate.** Infiltrated macrophages, resident Kupffer cells and RAW 264.7 cells (obtained from ATCC and free from mycoplasma) were stimulated with 200 µM palmitate conjugated with BSA or only BSA for one hour, and surface immune-fluorescence antibody staining was performed quickly for identification of macrophages. The cells were incubated with ROS Detection Reagents; 6-chloromethyl-2′, 7′-dichlorodihydrofluorescein diacetate, acetyl ester (CM-H$_2$DCF-DA from Invitrogen, Carlsbad, CA, USA) on 37 °C for 15 min, whereas incubated cells with PBS (Fluorescence Minus One; FMO) were used as negative control. This FMO control contained all the fluorochromes except for CM-H$_2$DCF-DA that was being measure, and the fluorescence was

slightly stronger than general unstained samples. Approximate fluorescence excitation is 492–495 nm and emission is 517–527 nm, so oxidation of these probes can be detected by monitoring FITC fluorescence with FACS LSRII. Histogram of relative fluorescence intensities was used to compare ROS generation.

**FACS analysis**. Cells were labeled with fluorescence tagged antibodies; under the anti-mouse CD16/CD32 (mouse Fc blocker, Clone 2.4G2) (BD Pharmingen, San Jose, CA, USA) and the Live/dead fixable aqua dead cell stain kit for 405 nm excitation (Life Technologies, Carlsbad, CA, USA). Infiltrated macrophages (CD11b$^+$, F4/80$^+$) and Kupffer cells (CD11b$^+$, F4/80$^{high}$) were described by using eFlour 450-conjugated anti-mouse CD45 (Clone 30-F11), FITC, PE, or PE/Cy7-F4/80 (Clone BM8) (eBioscience, San Diego, CA, USA), anti-mouse APC, APC-Cy7, or V500-CD11b (Clone M1/70) (BD Pharmingen, San Jose, CA, USA), and additional antibodies including anti-mouse FITC, PerCP-Cy5.5 or APC-Cy7-Ly-6C (Clone AL-21), and PE, PE-Cy7, or APC-Ly-6G (Clone 1A8) (BD bioscience, San Jose, CA, USA). Both cells were stained with anti-mouse APC-CD206 (MMR, Clone C068C2) (BioLegend, San Diego, CA, USA), PE or APC-conjugated TLR4/MD-2 (Clone MTS510), PerCP-Cy5.5-CD14 (Clone rmC5-3), PerCP-Cy5.5-MHC-II (I-A/I-E, Clone M5/114), PE-CD86 (Clone GL1), APC-CD11c (Clone HL3), FITC-CD1d (Clone 1B1) (BD Pharmingen, San Jose, CA, USA), anti-mouse FITC-CD80 (Clone 16-10A1) (eBioscience, San Diego, CA, USA), and compared expression of these markers. Lymphocytes (CD4 T, CD8 T, NK, NKT cells), Granulocytes (monocytes, neutrophils etc.) and regulatory T cells (CD4$^+$, CD25$^+$, FoxP3$^+$) were analyzed also through pseudo-color analysis plot. Stained cells read with FACS LSRII (BD Biosciences, San Jose, CA, USA), and the result was analyzed by FlowJo software (Flow Jo LLC, Ashland, OR, USA). Similarly, in human sample, infiltrated macrophages (CD14$^{high}$CD68$^{low}$) and resident macrophages (CD14$^{low}$CD68$^{high}$) were distinguished by using anti-human BV421 or V450-CD45 (Clone HI30) (BD Horizon, San Jose, CA, USA), anti-human PE or APC-Cy7-CD14 (Clone MΦP9) (BD Pharmingen, San Jose, CA, USA), and anti-human FITC or PE-Cy7-CD68 (Clone eBioY1/82A) (eBioscience, San Diego, CA, USA) under the human Fc blocker (BD Pharmingen, San Jose, CA, USA). Relative TLR4 expression (APC-conjugated CD284, Clone HTA125) (eBioscience, San Diego, CA, USA) was observed with histogram plots.

**Cell culture and palmitate treatment**. RAW 264.7 cells (immortalized mouse monocyte cells) were obtained from American Type Culture Collection (ATTC, Manassas, VA, USA) and were cultured in Dulbecco's modified Eagle medium (WG, Gyeongsan, Korea) supplemented with 10% fetal bovine serum (WG, Gyeongsan, South Korea) and 1% antibiotic-antimycotic (Thermo Fisher Scientific, Waltham, MA, USA) at 37 °C with 5% CO$_2$. Used cells were confirmed to be free from mycoplasma. For preparation of palmitate treatment, stock solution of 100 mM palmitate (Sigma-Aldrich, St. Louis, MO, USA) was dissolved with 0.1 M NaOH by heating at 70 °C in a hot water bath without vortex. 5% BSA solution with fatty acid free (Sigma-Aldrich, St. Louis, MO, USA) was prepared with H$_2$O and filtered through 0.22 μm syringe filter. By mixing two solutions, 5 mM palmitate-BSA was conjugated in a 55 °C water bath for 10–15 minutes, and cooling down before use. In some cases, 10 μM sulfosuccinimidyl oleate (SSO) (Cayman, Ann Arbor, MI, USA) or 80 μM dynasore (Sigma-Aldrich, St. Louis, MO, USA) was pre-treated for 10 min before palmitate treatment.

**Immunofluorescence staining**. For immunofluorescence staining, cells were plated in adhesive slides by using Cytospin centrifugation according to the manufacturer's instructions (Thermo Fisher Scientific, Waltham, MA, USA), and fixed with cold methanol (99%) for 10 min at 4 °C. After fixation, 0.1% Tween 20 (Sigma-Aldrich, St. Louis, MO, USA) was applied for permeabilization. Cells were incubated with primary antibodies of TLR4 (#293072) (Santa Cruz Biotechnology, CA, USA) and NOX2 (#ab31902) (Abcam, Cambridge, UK) for overnight at 4 °C and they were incubating with Alexa Fluor 488-conjugated anti-mouse IgG (#ab15109) or 594-conjugated anti-rabbit IgG (#ab150064) (Abcam, Cambridge, UK) for one hour at room temperature and mounted VECTASHIELD medium with DAPI (Vector laboratories, Burlingame, CA, USA) following the manufacturer's instructions. Fluorescence images were acquired using an Olympus BX51 microscope equipped with a CCD camera (Olympus, Tokyo, Japan) and computer-assisted image analysis with DP2-BSW.

**Ligand-dependent dimerization of hTLR4–MD2 complex**. Human TLR4 (amino acids 27-631) and protein A tagged MD2 at C-termini (amino acids 19–160) were co-expressed in High Five insect cells (Invitrogen, Carlsbad, CA, USA). The hTLR4–MD2–protein A complex was purified by IgG Sepharose affinity chromatography (GE Healthcare Life Sciences, Chicago, IL, USA) and the protein A tag was cleaved off with thrombin digestion on resin. The hTLR4–MD2 complex was subjected to gel filtration chromatography with Superdex 200 column (GE Healthcare Life Sciences, Chicago, IL, USA). hTLR4/MD2 complex (6.85 μM) was incubated with 68.5 μM of various ligands (LPS Ra, C16-BODIPY, Palmitate-BSA, BSA) for 3 h at 37 °C. The ligands were sonicated for 10 min prior to the reaction. The mixture of hTLR4–MD2 complex and each ligand was applied to Superdex 200 for gel filtration chromatography. The elution fractions were analyzed with SDS–PAGE followed by silver staining. Elution fractions of hTLR4–MD2 complex

with C16-BODIPY (Invitrogen, Carlsbad, CA, USA) were further analyzed by native PAGE in which the bands corresponding C16-BODIPY bound TLR4–MD2 were visualized with subsequent Coomassie (Thermo Fisher Scientific, Waltham, MA, USA) staining and illumination at 488 nm.

**Binding assay between palmitate and TLR4/MD2 protein**. Purified protein (mouse and human TLR4, MD2, TLR4–MD2 complex, CD14) was incubated with 200 μM fluorescence tagged palmitic acid (BODIPY) in room temperature for 30 min. The samples were loaded on gradient Native PAGE gel (4–15%) (Koma Biotech, Seoul, South Korea) at 4 °C.

**TUNEL staining**. Apoptotic cells were detected by *In situ* Cell Death Detection Kit (Roche, IN, USA) according to the manufacturer's instructions. In brief, depar-affinized liver sections were pre-treated with proteinase K (Sigma-Aldrich, St. Louis, MO, USA). The slide was incubated with TdT reaction solution in humidified chamber for 2 h and stop reaction. Positive cells were stained with FITC-Avidin D, and counter-staining was performed with DAPI. The number of positive cells was counted per equivalent area.

**Annexin V binding assay**. Early stage of apoptosis was analyzed by Annexin V and 7-Amino-Actinomycin (7-AAD) staining (BD Pharmingen, San Jose, CA, USA). After palmitate treatment for 1 h, macrophages and Kupffer cells were incubated with FITC-Annexin V and 7-AAD for 15 min in Annexin V binding buffer (BD Pharmingen, San Jose, CA, USA). Early apoptosis was represented as Annexin V positive and 7-AAD negative (Annexin V$^+$7-AAD$^-$).

**Human liver samples**. HCC tissue samples were obtained from the archives of the Department of Surgery, Chungnam National University Hospital (Daejeon, South Korea). HCC blood samples were obtained from the archives of the Division of Gastroenterology, Department of Internal Medicine, Chungnam National University Hospital. HCC tissues were subjected to fluorescence-activated cell sorting for CD68 and CD14 analyses. Peripheral blood mononuclear cells (PBMCs) from blood samples of healthy donors were isolated and analyzed. Purified CD14+ (Magnisort; affymetrix, Santa Clara, CA, USA) PBMCs were applied to ROS measurement after palmitate or palmitate treatment with dynasore pre-incubated or not. Authorization for the use of these tissues for research purposes and ethical approval were obtained from the Institutional Review Board of Chungnam National University Hospital (IRB number: 2016-03-02-003). Informed consents, which were approved by Institutional Review Board of Chungnam National University Hospital, were received from the entire patients who had provided the tissue or blood. Non-tumor lesions of liver tissue samples from a patient with fatty liver and eight patients with HCC ($n = 6$) or metastasis ($n = 2$) from colorectal cancer of stage IV were used in this research. All of these patients had undergone tumor resection operation between 2016 and 2017.

**Gene Expression Omnibus data set**. In order to analyze the expression level of genes in NASH liver, gene expression data were obtained from the National Center for Biotechnology Information (NCBI) Gene Expression Omnibus (GEO) database (accession code GSE63067) and previously published microarray data. The GSE63067 data set includes two human samples of steatosis and nine human samples for non-alcoholic steatohepatitis collected at the Swedish University of Agricultural Sciences, and it was used to define the molecular characteristics and to define non-alcoholic steatohepatitis early markers from Steatosis[52]. Nine NASH samples in the cohort were taken into consideration for analyses as compared with normal control.

**Statistical analysis**. All data are presented as the mean ± s.e.m. The statistical differences between groups were determined by the Student's t-test or one-way analysis of variance for multiple comparisons using GraphPad Prism 5 (GraphPad Software, La Jolla, CA, USA). Differences were considered statistically significant at $P < 0.05$.

**Data availability**. All relevant data that support the findings of this study are available from the corresponding author upon request.

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

## Acknowledgements

This work was supported by the National Research Foundation of Korea (NRF) grant funded by the Korean government (MSIP) (NFR-2015R1A2A1A10055551), the Korea Mouse Phenotyping Project (NRF-2014 M3A9D5A01073556) of the Ministry of Science, ICT and Future Planning through the National Research Foundation, and the Intelligent Synthetic Biology Center of Global Frontier Project funded by the Ministry of Science, ICT & Future Planning (2011-0031955), Republic of Korea.

## Author contributions

S.Y.K., J.-M.J., W.S., and W.-I.J. contributed to design and performance of experiments and analyses of data. M.-H.K., W.-M.C. J.-H.L., and Y.-R.S. contributed to isolation of mouse macrophages and flow cytometry analyses. W.Y. contributed to analyses of GEO database. S.Y.K., H.-S.Y. and Y.-S.L. contributed to isolation of human macrophages and

PBMCs from patients and to flow cytometry analyses. H.S.E., B.S.L., and K.C. contributed to collection of human liver samples and analyses of data. S.-J.K. and H.M.K. performed size exclusive chromatography and analyses of data. S.C.K., B.G., G.K., H.M.K., and W.-I.J. wrote the manuscript.

## Additional information

**Competing financial interests:** The authors declare no competing financial interests

