## [Peer Review File · Nature Communications]

Reviewers' comments:

Reviewer #1 (expert in steatosis and kupffer cells) (Remarks to the Author):

This study investigated the role of NOX2 in HFD-induced fatty liver. The authors discovered that palmitate binding to "monomeric" TLR4 complex, in contrast to LPS binding to dimeric TLR4 complex, induces ROS production, IL-1b and TNF induction, that are independent of MyD88 and TRIF. The authors first demonstrated NOX2 deficiency ameliorated HFD-induced hepatic steatosis in mice in vivo. Then, they showed inflammatory response and ROS production were decreased in CD11b+F4/80low macrophages in NOX2-deficient mice fed with high-fat diet for 12 weeks. CD11b+F4/80low macrophages had more M1 phenotype than resident CD11b+F4/80high Kupffer cells in WT livers. Interestingly, they found that palmitate treatment increased ROS generation in CD11b+F4/80low macrophages by the TLR4, NOX2 and dynamin-dependent manners, but not in Kupffer cells. They also found that endocytosis of palmitate/TLR4-MD2 complex is required for the generation of NOX2-mediated ROS that is independent of MyD88- and TRIF. Importantly, the study showed monomeric TLR4-MD2 complex interacted with palmitate. The study also used human cells and included NOX2 bone-marrow chimeric mouse experiment. The study was designed very carefully; the dataset is convincing, and the manuscript is well-written.

Major comments:

1. The study showed NOX2-mediated ROS contributes to pro-IL-1b and TNF α upregulation. The authors claimed this is the main mechanism. This upregulation is usually dependent on NF- κ B or JNK activation. Is NOX2 and ROS required for NF- κ B or JNK activation in response to LPS or palmitate?
2. Generally speaking, NADPH-oxidase-mediated ROS is associated with activation of inflammasome to secrete active IL-1b. This is recognized as the major mechanism of ROS-mediated inflammatory response in macrophages. Are NOX2 and ROS required for inflammasome activation in response to LPS or palmitate. The assessment of active form of caspase-1 and IL-1b should be provided. RAW cells may not be a suitable cell line to measure secreted IL-1b. In this case, this reviewer suggests to use bone marrow-derived macrophages.
3. Figure 5A: The authors claimed that there is no direct interaction between TLR4 and NOX2. What is the mechanism of TLR4-mediated NOX2 activation?
4. Figure 5C: The authors claimed that MyD88 and TRIF are not required for ROS production through silencing these genes by siRNA technology. Evidence of the silencing of MyD88 and TRIF has to be shown. Otherwise, the authors' argument is weak.
5. Figure 2F: Measurement of the number of apoptotic cells in whole livers. Are these cells either hepatocytes or non-parenchymal cells?
6. Figure 3D: iNOS and arginase-1 are typical M1 and M2 markers, respectively. However, These markers were not changed. The use of the concept M1-M2 polarization may be weak, and not related to the present study concept.

Minor comments:

1. Please provide how many animals and human samples the study used, and how many replicates the study performed for in vitro experiments in Figure legends.

Reviewer #2 (expert in TLR and immune response in liver) (Remarks to the Author):

This study describes that palmitate via TLR4 induces oxidative stress which results in fatty liver disease. I have a number of concerns.

1) These observations are not entirely novel. In fact there are numerous studies demonstrating 1) that inflammatory macrophage infiltrate liver in high fat systems and 2) that oxidants play a role. What may be somewhat new is the TLR4 dependent but Myd88 independent result however this needs to be verified in mice.

2) Please demonstrate that a similar finding as in NOX2 mice can be obtained in TLR4 deficient mice preferably on a LyzM promoter.

3) please demonstrate that MyD88^{-/-} mice do not have the same phenotype as TLR4^{-/-} mice in these models.

4) It is not clear what the bone marrow transplant data demonstrate. NOX2 is primarily made by macrophage including Kupffer cells and neutrophils. Using bone marrow transplant does not segregate these different cells.

5) The human data while a nice addition to this paper have previously been documented and should be cited. There is a nice review by Tacke and Krenkel in Nature Reviews Immunology 2017 that summarize much of that human work.

Reviewer #3 (expert in NASH, liver metabolism, TLR) (Remarks to the Author):

In this manuscript, the authors reported that the saturated fatty acid palmitate stimulates infiltrated hepatic macrophages to generate reactive oxygen species (ROS) via dynamin-mediated endocytosis of TLR4 and NOX2, independent of MYD88 and TRIF. Interestingly, this study demonstrated that palmitate binds to a monomeric TLR4-MD2 complex, unlike LPS-mediated dimerization of the TLR4-MD2 complex, to trigger ROS generation and macrophage inflammation. The in vitro analyses suggested that endocytosis of TLR4 and NOX2 into macrophages might be a therapeutic target for NAFLD. Together, the findings from this study provide novel insights into the understanding of and potential therapeutic intervention for NAFLD. However, there are some major concerns on this work:

1. Regarding the causes of ROS reduction in NOX2 KO livers under the HFD, is this attributed to the reduced numbers of infiltrated CD11b⁺F4/80 low macrophages (as the authors showed that the population of infiltrated CD11b⁺F4/80 low macrophages was significantly decreased in the absence of NOX2)? Furthermore, is the ROS reduction by CD11b⁺F4/80 low macrophages associated with macrophage-specific cell death events?

2. Does NOX2 deficiency decrease inflammatory responses and ROS production in CD11b⁺F4/80 high Kupffer cells? In another word, does NOX2 deficiency generally decrease ROS and inflammatory responses in most cell types? The related results and conclusions need to be addressed in the manuscript.

3. To conclude direct binding of palmitate to the TLR4-MD2 complex, the authors performed PAGE analysis with BODIPY-labeled fatty acid analogues (C16-BODIPY) and purified TLR4-MD2 protein. However, the binding assay with artificial C16-BODIPY is not sufficient to conclude direct binding of palmitate to the TLR4-MD2 complex. Gel super-shift assay with radioactive isotope-labeled palmitate is required to confirm direct binding of palmitate to the TLR4-MD complex in the context of NOX2-mediated ROS generation in macrophages.

Minor concerns:

There are few typos in the manuscript:

1. Page 4, line 85, "He we report that in CD11b+F4/80low hepatic macrophages,"
2. Page 5, line 109, "These differences were evident at early as 6 weeks of HFD"

Reviewers' comments:

Reviewer #1 (expert in steatosis and kupffer cells) (Remarks to the Author):

This study investigated the role of NOX2 in HFD-induced fatty liver. The authors discovered that palmitate binding to "monomeric" TLR4 complex, in contrast to LPS binding to dimeric TLR4 complex, induces ROS production, IL-1b and TNF induction, that are independent of MyD88 and TRIF. The authors first demonstrated NOX2 deficiency ameliorated HFD-induced hepatic steatosis in mice in vivo. Then, they showed inflammatory response and ROS production were decreased in CD11b+F4/80low macrophages in NOX2-deficient mice fed with high-fat diet for 12 weeks. CD11b+F4/80low macrophages had more M1 phenotype than resident CD11b+F4/80high Kupffer cells in WT livers. Interestingly, they found that palmitate treatment increased ROS generation in CD11b+F4/80low macrophages by the TLR4, NOX2 and dynamin-dependent manners, but not in Kupffer cells. They also found that endocytosis of palmitate/TLR4-MD2 complex is required for the generation of NOX2-mediated ROS that is independent of MyD88- and TRIF. Importantly, the study showed monomeric TLR4-MD2 complex interacted with palmitate. The study also used human cells and included NOX2 bone-marrow chimeric mouse experiment. The study was designed very carefully; the dataset is convincing, and the manuscript is well-written.

We thank the reviewer for the positive comments.

Major comments:

1. The study showed NOX2-mediated ROS contributes to pro-IL-1b and TNFa upregulation. The authors claimed this is the main mechanism. This upregulation is usually dependent on NF-kB or JNK activation. Is NOX2 and ROS required for NF-kB or JNK activation in response to LPS or palmitate?

Answer: To address this important question, we performed additional experiments and added data on Fig. 5a and Supplementary Fig. 4c. Similar to earlier findings by others (Shi et al., J Clin Invest 2006, 116:3015-3025 and Nat Commun 2017, 3;8:13997), palmitate treatment induced phosphorylation of both NF-kB and JNK, while NOX2 depletion abolished palmitate-mediated activation of NF-kB and JNK. These data suggest that activation of NF-kB and JNK is dependent on ROS by palmitate-mediated NOX2. These findings are described on page 7 and discussed on page 13.

2. Generally speaking, NADPH-oxidase-mediated ROS is associated with activation of inflammasome to secrete active IL-1b. This is recognized as the major mechanism of ROS-mediated inflammatory response in macrophages. Are NOX2 and ROS required for inflammasome activation in response to LPS or palmitate. The assessment of active form of caspase-1 and IL-1b should be provided. RAW cells may not be a suitable cell line to measure secreted IL-1b. In this case, this reviewer suggests to use bone marrow-derived macrophages.

Answer: Thanks for this constructive comment. It has been previously reported that RAW 264.7 cells do not produce mature form of IL-1 β due to lack of ASC (apoptosis-associated speck-like protein containing a C-terminal caspase-activating recruiting domain) (Pelegrin P, Barroso-Gutierrez C, Surprenant A. P2X7 receptor differentially couples to distinct release pathways for IL-1beta in mouse macrophage. J Immunol 2008, 180:7147-7157). Thus, we treated BMDMs of WT and NOX2 KO mice with palmitate. Similar to findings of RAW 264.7 cells, palmitate treatment not only increased the expression of IL-1 β protein, but also activated caspase-1 in BMDMs of WT mice, but not of NOX2 KO mice (Fig. 5b,c). These findings are described on

page 7 and discussed on page 13 in the manuscript.

3. Figure 5A: The authors claimed that there is no direct interaction between TLR4 and NOX2. What is the mechanism of TLR4-mediated NOX2 activation?

Answer: Unfortunately, this is still unclear, but we have now made the following discussion in the Discussion. According to previous papers (Reference 38-40), in response to diverse stimuli, clustering of lipid rafts (LRs) in the plasma membrane drives NOX assembly and aggregation, and LPS also stimulates association of TLR4 and its accessory proteins with LRs, triggering a signal cascade. These findings suggest that TLR4 bound by palmitate might induce clustering of LRs, which are also enriched in components of the NOX2 complex, leading to the internalization of these molecules to generate ROS. However, further studies are required to elucidate the exact mechanism involved. We tried to address this in discussion (first paragraph at page 13).

4. Figure 5C: The authors claimed that MyD88 and TRIF are not required for ROS production through silencing these genes by siRNA technology. Evidence of the silencing of MyD88 and TRIF has to be shown. Otherwise, the authors' argument is weak.

Answer: Thank you for bringing this to our attention. We added evidence of silencing on MyD88 and TRIF in Supplementary Fig 4d.

5. Figure 2F: Measurement of the number of apoptotic cells in whole livers. Are these cells either hepatocytes or non-parenchymal cells?

Answer: We added apoptotic images to Fig 2f. Although the nature of apoptotic cells is unclear, they may be hepatocytes, as suggested by the reduced levels of ALT and AST in NOX2 KO mice compared with WT mice (Fig. 1c). In addition, there were no direct induction of apoptosis and the ratio of apoptotic CD11b⁺F4/80^{low} macrophages in response to *in vitro* palmitate treatment was similar between WT and NOX2 KO mice (Fig. 2g). This suggests that the lower frequency of CD11b⁺F4/80^{low} macrophages in livers of NOX2 KO mice (Fig. 2b) might be due to weaker inflammatory responses and less severe hepatocyte injury than in WT mice, using a high-fat diet-induced hepatic steatosis model. These findings are described on page 6.

6. Figure 3D: iNOS and arginase-1 are typical M1 and M2 markers, respectively. However, These markers were not changed. The use of the concept M1-M2 polarization may be weak, and not related to the present study concept.

Answer: I agree with your suggestion. As you indicated, we replaced M1 phenotype of macrophages with proinflammatory phenotype of macrophages.

Minor comments:

1. Please provide how many animals and human samples the study used, and how many replicates the study performed for in vitro experiments in Figure legends.

Answer: we described them in the materials and methods and figure legends as well.

Reviewer #2 (expert in TLR and immune response in liver) (Remarks to the Author):

This study describes that palmitate via TLR4 induces oxidative stress which results in fatty liver disease. I have a number of concerns.

1) These observations are not entirely novel. In fact there are numerous studies demonstrating 1) that inflammatory macrophage infiltrate liver in high fat systems and 2) that oxidants play a role. What may be somewhat new is the TLR4 dependent but Myd88 independent result however this needs to be verified in mice.

Answer: Thanks for your constructive comments. To more clearly demonstrate whether or not palmitate-mediated ROS generation is independent of MYD88 and TRIF, we performed functional validation *in vivo*. We perfused the liver via the portal vein with C16-BODIPY or palmitate in WT, TLR4 KO, NOX2 KO and MYD88/TRIF double KO mice (Fig. 5e), and demonstrated that the uptake of C16-BODIPY by macrophages, Kupffer cells and neutrophils. However, only CD11b⁺F4/80^{low} macrophages generated ROS in response to exogenous palmitate, although NOX2 was highly expressed in macrophages, Kupffer cells and neutrophils (Fig. 5f and Supplementary Fig. 4e). More importantly, TLR4 and NOX2 were indispensable for palmitate-induced ROS generation, whereas MYD88 and TRIF were not required. These findings support our hypothesis and were addressed in the Discussion (pages 11-13)

2) Please demonstrate that a similar finding as in NOX2 mice can be obtained in TLR4 deficient mice preferably on a LyzM promoter.

Answer: This is a great suggestion. Unfortunately, we could not generate macrophage-specific TLR4 KO mice as we don't have access to TLR4-floxed mice. According to data in the literature, less fat accumulation is expected in high-fat fed TLR4 KO compared to WT mice due to reduced inflammatory responses via the LPS-TLR4-MyD88 or LPS-TLR4-TRIF pathway in macrophages. We found that freshly isolated macrophages of TLR4 KO mice or RAW 264.7 cells with siRNA-mediated suppression of TLR4 did not generate ROS after palmitate treatment *in vitro* and these results were comparable with those obtained in NOX2 KO macrophages (Fig. 4a and Fig. 5d). Indeed, we confirmed suppressed ROS generation in hepatic macrophages of TLR4 KO and NOX2 KO mice compared with WT mice in response to perfusing the liver with palmitate (Fig. 5g). Based on *in vitro* and *in vivo* experiments, we could predict similar findings in TLR4 KO mice compared with NOX2 KO mice. We described these findings and added data in the manuscript.

3) please demonstrate that MyD88^{-/-} mice do not have the same phenotype as TLR4^{-/-} mice in these models.

Answer: When LPS levels are elevated in the portal circulation in response to high-fat diet feeding, it is not easy to distinguish the nature if the difference between TLR4 KO and MyD88 KO mice. Instead, using intraportal palmitate infusion (Fig. 5e) or *in vitro* palmitate treatment of primary macrophages from MYD88/TRIF double KO mice or RAW 264.7 cells with deletion of MYD88 or TRIF (Fig. 5d), we found increased ROS generation compared to macrophages of TLR4 KO mice or TLR4-depleted RAW 264.7 cells. Similarly, one study demonstrated that treatment with minimally oxidized LDL generated NOX2-mediated ROS in peritoneal

macrophages in a TLR4-dependent, but MYD88-independent manner, supporting our data. We added these findings and this reference in the result and discussion (page 12).

4) It is not clear what the bone marrow transplant data demonstrate. NOX2 is primarily made by macrophage including Kupffer cells and neutrophils. Using bone marrow transplant does not segregate these different cells.

Answer: we fully agree, as it is difficult to delete NOX2 gene only in CD11b⁺F4/80^{low} macrophages while sparing Kupffer cells and neutrophils. Expression of the lysozyme 2 gene (Lyz2 or LyzM) is detectable both in macrophages and neutrophils, so using LyzM cre mice is not suitable for achieving macrophage-specific gene deletion. Although the data from chimeric mice generated by bone marrow transplantation (BMT) do not establish the distinct roles of macrophages, Kupffer cells and neutrophils, other findings do. First, a recent paper that you recommended to us reported that the population of Kupffer cells is maintained by self-renewal rather than by trans-differentiation of infiltrating monocytes, indicating that BMT might affect the latter, but not Kupffer cells. Second, we found that CD11b⁺Ly6G⁺ neutrophils did not generate ROS in response to *in vivo* treatment with the palmitate-mimic C16-BODIPY (Fig. 5f). This suggests that palmitate-mediated, NOX2-dependent ROS generation is limited to CD11b⁺F4/80^{low} macrophages. Furthermore, using a similar BMT approach, another study demonstrated that WT mice with NOX2-deficient bone marrow had decreased ROS production in mononuclear cells and impaired neovascularization following hindlimb ischemia. Therefore, BMT may be useful method in our study. We described this and added a related reference (#29) to the discussion (page 11).

5) The human data while a nice addition to this paper have previously been documented and should be cited. There is a nice review by Tacke and Krenkel in Nature Reviews Immunology 2017 that summarize much of that human work.

Answer: As you recommended, we added this reference (#4) on our manuscript.

Reviewer #3 (expert in NASH, liver metabolism, TLR) (Remarks to the Author):

In this manuscript, the authors reported that the saturated fatty acid palmitate stimulates infiltrated hepatic macrophages to generate reactive oxygen species (ROS) via dynamin-mediated endocytosis of TLR4 and NOX2, independent of MYD88 and TRIF. Interestingly, this study demonstrated that palmitate binds to a monomeric TLR4-MD2 complex, unlike LPS-mediated dimerization of the TLR4-MD2 complex, to trigger ROS generation and macrophage inflammation. The *in vitro* analyses suggested that endocytosis of TLR4 and NOX2 into macrophages might be a therapeutic target for NAFLD. Together, the findings from this study provide novel insights into the understanding of and potential therapeutic intervention for NAFLD. However, there are some major concerns on this work:

1. Regarding the causes of ROS reduction in NOX2 KO livers under the HFD, is this attributed to the reduced numbers of infiltrated CD11b⁺F4/80^{low} macrophages (as the authors showed that the population of infiltrated CD11b⁺F4/80^{low} macrophages was significantly decreased in the absence of NOX2)? Furthermore, is the ROS reduction by CD11b⁺F4/80^{low} macrophages associated with macrophage-specific cell death events?

Answer: To answer these important questions, we performed additional experiments, which showed that *in vitro* palmitate treatment did not cause death of CD11b⁺F4/80^{low} macrophages from WT or NOX2 KO mice (Fig. 2g). Also, we feel that the decreased population of CD11b⁺F4/80^{low} macrophages might be a consequence of less inflammation in NOX2 KO mice. We added these findings and updated the discussed accordingly.

2. Does NOX2 deficiency decrease inflammatory responses and ROS production in CD11b⁺F4/80^{high} Kupffer cells? In another word, does NOX2 deficiency generally decrease ROS and inflammatory responses in most cell types? The related results and conclusions need to be addressed in the manuscript.

Answer: We agree that this is an important issue. We found that NOX2 gene expression was high in CD11b⁺F4/80^{low} macrophages, CD11b⁺F4/80^{high} Kupffer cells and CD11b⁺Ly6G⁺ neutrophils compared to other cells such as hepatocytes, hepatic stellate cells and liver sinusoidal endothelial cells (Supplementary Fig. 4e). In response to intra-portal infusion of palmitate, CD11b⁺F4/80^{low} macrophages isolated from the liver was the only cell type generating increased amount of ROS, whereas there was no change of ROS generation in CD11b⁺F4/80^{high} Kupffer cells or CD11b⁺Ly6G⁺ neutrophils (Fig. 5f). This suggest that the CD11b⁺F4/80^{low} macrophage is the major source of NOX2-mediated ROS generation by palmitate. We added and discussed these data in the manuscript.

3. To conclude direct binding of palmitate to the TLR4-MD2 complex, the authors performed PAGE analysis with BODIPY-labeled fatty acid analogues (C16-BODIPY) and purified TLR4-MD2 protein. However, the binding assay with artificial C16-BODIPY is not sufficient to conclude direct binding of palmitate to the TLR4-MD2 complex. Gel super-shift assay with radioactive isotope-labeled palmitate is required to confirm direct binding of palmitate to the TLR4-MD complex in the context of NOX2-mediated ROS generation in macrophages.

Answer: We fully agree. Indeed, a recent paper clearly showed that direct binding of palmitate with MD2 (Saturated palmitic acid induces myocardial inflammatory injuries through direct binding to TLR4 accessory protein MD2. Nature communication 2017 Jan 3;8:13997). We described it in the discussion (page 12) and added this paper as a reference (#37).

Minor concerns:

There are few typos in the manuscript:

1. Page 4, line 85, “He we report that in CD11b⁺F4/80^{low} hepatic macrophages,”

Answer: Thank you for bringing this to our attention. It was corrected as indicated.

2. Page 5, line 109, “These differences were evident at early as 6 weeks of HFD”

Answer: Thank you for bringing this to our attention. It was corrected as indicated.

REVIEWERS' COMMENTS:

Reviewer #1 (Remarks to the Author):

The authors addressed the reviewers' comments accordingly. No further comments.

Reviewer #2 (Remarks to the Author):

Good rebuttal

Reviewer #3 (Remarks to the Author):

As the comment raised in the first round, it is important to confirm direct binding of palmitate to the TLR4-MD complex in the context of NOX2-mediated ROS production, since palmitate binding to the TLR4-MD complex is a major mechanistic basis of the findings presented in this study. Although the author cited a recent report showing the direct binding of palmitate to MD2, it looks still necessary to confirm that the binding of palmitate to TLR4-MD complex is coincident with ROS production in macrophages.

REVIEWERS' COMMENTS:

Reviewer #1 (Remarks to the Author):

The authors addressed the reviewers' comments accordingly. No further comments.

Reviewer #2 (Remarks to the Author):

Good rebuttal

Reviewer #3 (Remarks to the Author):

As the comment raised in the first round, it is important to confirm direct binding of palmitate to the TLR4-MD complex in the context of NOX2-mediated ROS production, since palmitate binding to the TLR4-MD complex is a major mechanistic basis of the findings presented in this study. Although the author cited a recent report showing the direct binding of palmitate to MD2, it looks still necessary to confirm that the binding of palmitate to TLR4-MD complex is coincident with ROS production in macrophages.

Answer

Thank you for your comments. In parallel with a previous study, we first demonstrated direct binding of palmitic acid combined with fluorescence (C16-BODIPY) to TLR4-MD2 and MD2 proteins of both mouse and human in Figure 6C, Supplementary Figure 5 and Supplementary Figure 6b. Second, in contrast to LPS, we further showed monomeric interaction of TLR4-MD2 in response to palmitic acid in Figure 6c and 6d. Lastly, we identified that NOX2-mediated ROS generation was a TLR4/NOX2-dependent but MYD88/TRIF-independent manner in response to palmitic acid using siRNA transfection *in vitro* and direct circulation of palmitate to *Myd88/Trif* double KO mice *in vivo* (Figure 5d-g). These findings may accommodate reviewer's concerns.